# MACRO: A Multi-Head Attentional Convolutional Recurrent Network for the Classification of Co-Occurring Diseases in 12-Lead ECGs

Vanessa Borst (iD) (✉), Robert Leppich (iD) and Samuel Kounev (iD)
University of Würzburg, Würzburg, Germany
Email: {vanessa.borst, robert.leppich, samuel.kounev}@uni-wuerzburg.de

*Abstract*—Cardiovascular diseases (CVDs) are a significant global health concern, causing more deaths than several types of cancer combined. Early detection and proper treatment are crucial for better clinical outcomes. Electrocardiograms (ECGs) offer valuable insights into the presence of CVDs. However, extracting powerful features from raw ECG signals for reliable automated diagnostics remains challenging due to high inter- and intra-patient variability, diversity of rhythmic and morphological abnormalities, and noise distributions. This paper proposes a deep learning architecture for the automated detection of diseases in 12-lead ECG data, capable of recognizing concurrent irregularities through multi-label classification. Moreover, we present a novel method that combines deep feature extraction with binary machine learning classifiers. To account for the distinct characteristics of various ECG leads, we use a multi-loss optimization strategy. Our methodology is rigorously evaluated through 10-fold cross-validation using the publicly available CPSC 2018 dataset. With macro $F_1$ and AUC scores reaching up to 85.2% and 98.0%, respectively, our approach demonstrates advantage over existing state-of-the-art methods. At the same time, our architecture remains lightweight with approximately 1.7 million trainable parameters, which represents a reduction in the number of parameters of up to 68% compared to previous methods. Assessing the generalizability of our approach, we further evaluated it on the PTB-XL dataset and achieved macro $F_1$, AUC, and accuracy scores comparable to existing methods, demonstrating the robustness of our model across diverse datasets. This advancement holds promise for enhanced automated diagnosis and improved patient care in the context of CVDs. Our code is available at: https://github.com/VanessaBorst/MACRO.

*Index Terms*—ECG signal, Ensemble, Gradient boosting, Healthcare, Multi-head attention, Multi-label classification.

## I. INTRODUCTION

Cardiovascular disease (CVD) is a leading cause of death worldwide, accounting for approximately 17.9 million deaths annually according to the World Health Organization [1]. Timely diagnosis is crucial, and the electrocardiogram (ECG) has become a vital, non-invasive, and cost-effective modality for detecting CVD by capturing the heart's electrical activity through strategically placed body electrodes that convert it into a distinctive waveform. Deviations in these waveform patterns serve as indicative markers for the presence of CVD [2]. In clinical practice, the 12-lead ECG, consisting of the six limb leads (I, II, III, aVL, aVR, and aVF) and the six precordial or chest leads (V1, V2, V3, V4, V5, and V6), provides different perspectives of the heart's electrical activity by recording it from distinct angles [3]. As a result, different leads incorporate lead-specific characteristics. However, automated detection of

irregularities faces challenges that cause significant misdiagnosis in computerized interpretation of the ECG (CIE), even with existing commercial algorithms [4]. These challenges include inter- and intra-patient variability, common features leading to similar ECG findings across different cardiac abnormalities, co-occurrence of multiple irregularities, and artefacts introduced during acquisition.

Despite these challenges, researchers have proposed a plethora of techniques for CVD detection, including traditional statistical and machine learning (ML) methods [5], [6]. However, these methods typically require complex preprocessing and feature extraction (FE) prior to detection [5]. As a result, deep learning (DL) paradigms have emerged as end-to-end solutions that eliminate the need for extensive preprocessing and labor-intensive manual feature engineering. Many of these combine FE and classification [7], [8], where some process raw ECG data directly [9] or only require minimal modifications like padding [10]. Convolutional Neural Networks (CNNs) have been extensively explored for ECG classification at both the heartbeat level [11] and in methods processing ECG sequences [8]. While CNNs excel at extracting discriminative features from spatially and locally related data, they often neglect the temporal properties of ECG signals [12]. In contrast, Recurrent Neural Networks (RNNs), such as LSTMs and GRUs, are designed for handling sequential data of varying lengths but lack local information [12]. Capturing temporal dynamics inherent in ECGs, RNNs have been applied to both heartbeat [13] and sequence-based [14] classification. In addition, hybrid approaches stacking CNNs and RNNs [7], [15] have proven successful, leveraging the strengths of both to extract local features and aggregate them along time [12].

Mostly within the last five years, advanced deep learning concepts have been adopted for 12-lead CVD detection, often inspired by breakthroughs in domains such as computer vision [16] or natural language processing [17]. Such adoptions include residual neural networks [18], [19], various types of attention mechanisms [10], [20], [21], and transformer models [22]. Residual networks (ResNets) exhibit numerous variants, such as the integration of handcrafted (expert) features [23] or their combination with RNNs [24]. ResNets with attention modules, such as squeeze-and-excitation (SE) blocks [25] or convolutional block attention (CBAM) modules [26], and methods that merge (SE)ResNets with transformer variants [27], [28] have also been successfully applied.

Lastly, approaches that process different ECG leads separately [21], [29], [30] or in groups [31] by dedicated feature extraction components per lead (group) within different network branches have made notable contributions. However, despite recent advances, certain limitations remain:

1) **Expert-Driven FE and Limited Temporal Scope of ML** Traditional ML approaches are less data-intensive and therefore well-suited for domains with limited labeled data, like medicine. However, they often require expert-crafted features, such as RR intervals, rather than automatically learning features from the data [2]. Moreover, existing methods frequently restrict their scope to single heartbeats [5] or short windows [6], thus failing to account for the dynamic ECG evolution over time.

2) **Large Model Size of DL Methods:** Despite their considerable success and end-to-end training capabilites, the model size of advanced deep learning architectures, such as Transformer [17], continues to grow. This not only results in enormous training costs and increased hardware requirements, but also impedes their applicability in hardware-constrained real-world scenarios.

3) **Independent Lead Analysis:** Most methods concatenate all ECG channels into a single 2D input matrix. Although DL models can learn to extract useful features from multivariate data implicitly or through network internals such as attention [20], without special operations (e.g., group convolution), all leads are initially treated equally and as a whole [27], [28]. However, from a medical perspective, certain heart conditions, such as atrial fibrillation (AF), show characteristic patterns in one or a few rather than in all leads [3]. Given that different leads can contribute differently to CVD detection, the ability of a model to capture comprehensive diagnostic information may be improved by learning class-specific features from the respective leads. However, most techniques lack mechanisms for extracting lead-specific features per class, such as analyzing leads separately in dedicated branches.

To address these limitations, we introduce a multi-branch (MB) architecture that independently extracts features from all twelve ECG leads and then fuses them into a comprehensive representation. Each branch, denoted as BranchNet, processes a single lead with a shared structure. Leveraging the concept of lead-specific features, we employ our MB network as a feature extractor coupled with individual binary classifiers for each label. This allows the models to assign varying importance to distinct lead-specific features based on the specific class. In summary, our key contributions are as follows:

1) **Multi-Branch Framework with Lead-Individual FE**: We introduce MACRO, a **M**ulti-Head **A**ttentional **C**onvolutional **R**ecurrent Network for the classification of Co-**O**ccurring diseases in 12-Lead ECGs, and Multi-Branch MACRO (MB-M). MB-M uses the structure of MACRO in twelve specialized BranchNets, each focused on extracting features from a single ECG lead. Then, it unifies the intermediate feature maps of these branches for a holistic understanding of cardiac activity.

2) **Novel Fusion of Deep FE and ML Ensemble**: In order to combine the strengths of ML and DL, we repurpose MB-M as deep feature extractor and integrate it with an ensemble of binary classifiers, training dedicated gradient boosting models per class.

3) **Detailed Model Size and Performance Evaluation**: We thoroughly evaluate our framework through 10-fold cross-validation on the public CPSC 2018 benchmark dataset, where it outperforms existing methods while maintaining a small parameter count. Ablation studies explore the impact of specific network components.

The remainder of this paper is structured as follows. Section II details the MACRO architecture, its multi-branch variant MB-M, and the gradient boosting classifiers. Section III outlines the experimental setup. Finally, we report and analyze results in Section IV, and we draw conclusions in Section V.

## II. METHODOLOGY

To address co-occurring diseases, we formulate the problem as a 12-lead multi-label classification task. Our models process ECG recordings $x \in \mathbb{R}^{L \times 12}$ of length $L$. Their goal is to predict one or more classes from a pool of $C$ potential classes for each recording concurrently.

### A. The MACRO Architecture

As summarized in Fig. 1, our MACRO model comprises a convolutional feature extraction module, a bidirectional gated recurrent unit (BiGRU), a multi-head attention mechanism, and a classification head. Inspired by Chen et al. [10], the winners of the China Physiological Signal Challenge (CPSC) 2018, it shares a similar overall structure with the 130 models constituting their ensemble. However, we do not adhere to their ensemble strategy but utilize MACRO as a standalone model. Furthermore, we introduce substantial architectural modifications, such as refining the CNN module with features like residual connections and varying channel amounts, and incorporating multi-head attention. Chen et al.'s architecture serves as a baseline in our study.

*1) Convolutional Feature Extraction:* In the first submodule of MACRO, a stack of five blocks is employed (cf. [10]), as illustrated in orange in Fig. 1. Each contains two 1D convolutional layers with kernel size three and stride one, followed by a downsampling operation that is realized as 1D convolution with stride two. The downsampling layers have a kernel size of 24 in the first four CNN blocks and 48 in the last block. Non-linear transformations are applied using leaky rectified linear units (LeakyReLU) with a negative slope value of 0.3, and each block concludes with a dropout layer set at a rate of 0.2.

Compared to Chen et al.'s model, we make several beneficial alterations based on preliminary experiments: First, we introduce skip connections [16] within all network blocks, implementing the pre-activation design paradigm [32]. Linear projections, utilizing dedicated $(1 \times 1)$ convolutional layers with a stride of two, are employed to align spatial and channel dimensions. Second, an initial block with a single 1D convolutional layer with a kernel size of 16 is added. Third,

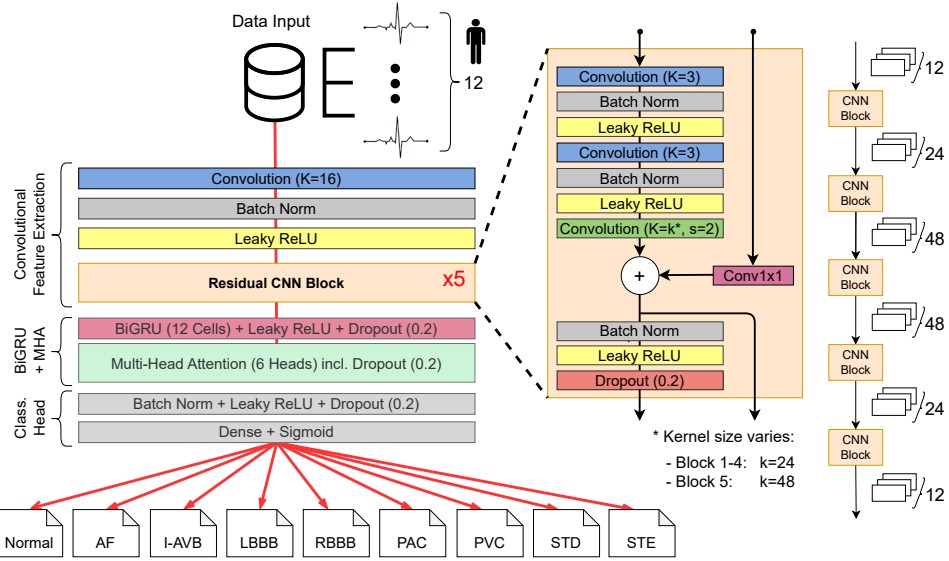

Fig. 1: Overview of the proposed MACRO architecture

we vary the amount of channels across blocks (cf. right side of Fig. 1), and finally, we incorporate batch normalization into the CNN module. The introduced skip connections, aligned with the pre-activation design, establish a "clean" pathway, enhancing information propagation and facilitating the direct flow of gradients. The additional initial block prevents the transfer of raw signals through shortcuts, bypassing any normalization or scaling. The progressive increase in the count of feature maps within the first half of the CNN module and subsequent reduction in the second half aim to first enrich and later compress knowledge for the comparatively small BiGRU module that follows. Lastly, the inclusion of batch normalization within the CNN enhances training effectiveness.

*2) Bidirectional Gated Recurrent Unit (BiGRU):* The CNN output feature map, denoted as $f_{CNN}$, is fed into a BiGRU layer with 12 units. This BiGRU interprets $f_{CNN}$ as a time series of length $T$, processing it in both forward and backward directions. At any time step $t$, the forward GRU aggregates information from time steps 1 to $t$, and the backward GRU gathers information from time steps $T$ to $t$. Combining both, the forward hidden state $h_t^{(f)} \in \mathbb{R}^{12}$ and the backward hidden state $h_t^{(b)} \in \mathbb{R}^{12}$ are concatenated into a single vector, denoted as $h_t^{BiRNN} = [h_t^{(f)}, h_t^{(b)}] \in \mathbb{R}^{24}$, encapsulating contextual information surrounding the input $x_t$ at time step $t$. Finally, these concatenated vectors undergo a LeakyReLU activation function and a dropout layer with a rate of 0.2.

*3) Multi-Head Attention (MHA):* The multi-head attention (MHA) layer of MACRO determines the importance weights for the BiGRU hidden states across different time steps. It combines the 24-dimensional input vectors $h_t^{BiRNN}$ of all time steps into a single 24-dimensional output vector using a weighted sum, providing a comprehensive data representation. While our reimplemented baseline model uses Chen et al.'s attention mechanism, we will discuss the MHA of MACRO next. A detailed visualization is available in Fig. 2.

In a MHA mechanism with $h$ heads, the attention function is computed independently $h$ times in parallel [17]. Each head $h_i$ has its own linear projection matrices $W_i^{(Q)} \in \mathbb{R}^{d_{model} \times d_q}$, $W_i^{(K)} \in \mathbb{R}^{d_{model} \times d_k}$, and $W_i^{(V)} \in \mathbb{R}^{d_{model} \times d_v}$, which are used to project the $d_{model}$-dimensional queries, keys, and values to $d_q, d_k$ and $d_v$ dimensions, respectively. In this work, all dimensions are set to be the same, specifically $d_q = d_k = d_v = d_{model}/h$ with $d_{model} = 24$. For each time step $t$, the BiGRU output vector $h_t^{BiRNN}$ is used as both the key and value. Differing from the self-attention used in Transformers [17], the query $q$ is initialized as a random vector and jointly learned. We apply a $tanh$ activation function after each head's linear key transformation through $W_i^{(K)}$ to maintain the non-linearity within key derivation, similar to the attention mechanism in our baseline.

In practice, the attention function is computed over all $T$ time steps simultaneously. The keys and values $h_t^{BiRNN}$ for all time steps $t \in 1, ..., T$ are combined into matrices $K \in \mathbb{R}^{T \times d_{model}}$ and $V \in \mathbb{R}^{T \times d_{model}}$. The query $q \in \mathbb{R}^{d_{model}}$ remains a single vector used across all time steps. Thus, the calculation for head $h_i$ can be summarized as follows, producing a vector of dimension $d_v$:

$$h_i = \text{Attention}\left(q\, W_i^{(Q)}, K\, W_i^{(K)}, V\, W_i^{(V)}\right) \quad (1)$$

$$= \text{Attention}(q_i, K_i, V_i) \quad (2)$$

$$= 1.5\text{-Entmax}\left(\frac{q_i K_i^T}{\sqrt{d_k}}\right) V_i \quad (3)$$

Unlike the non-scaled dot-product in the baseline attention mechanism, the MHA layer uses a scaled dot-product. We also employ the $\alpha$-entmax function [33] with $\alpha = 1.5$ instead of softmax, producing sparse probability distributions for weight determination. Moreover, a dropout with a rate of 0.2 is applied to the Entmax scores to prevent overfitting (cf. Fig. 2).

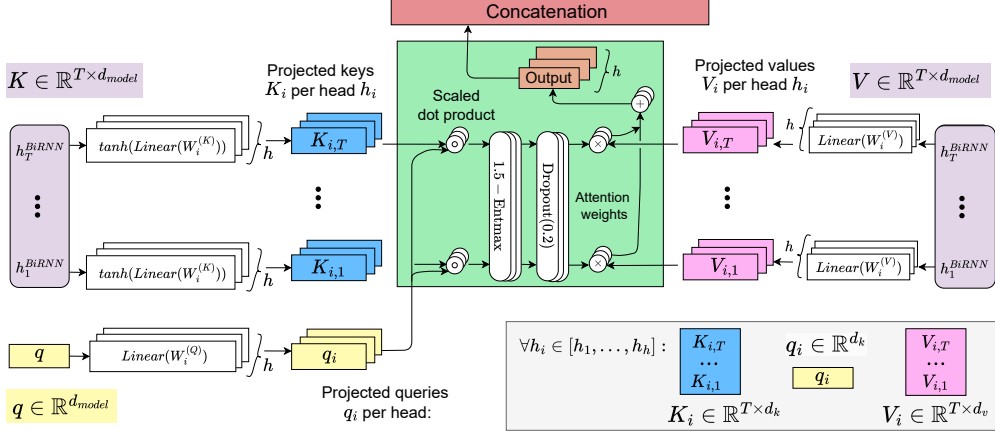

Fig. 2: Detailed view of our MHA with $\alpha - Entmax$ from a zoomed-in perspective.

Lastly, the outcomes from all $h$ heads are concatenated into a vector of dimension $h \cdot d_v$. It undergoes further processing through a single-layer MLP with weight matrix $W^O \in \mathbb{R}^{h \cdot d_v \times d_{model}}$, producing a result vector of dimension $d_{model}$ as follows:

$$MultiHead(q, K, V) = \text{concat}(h_1, ..., h_h) W^O \qquad (4)$$

This MHA mechanism enables simultaneous attention to diverse parts of the input sequence, augmenting the model's capacity to capture complex relations.

*4) Classification Head:* After the MHA layer, its output undergoes further transformations: batch normalization for scaling, LeakyReLU activation for non-linearity, dropout with rate 0.2 for regularization, and finally, a dense layer with a Sigmoid activation function. This dense layer maps the inputs to a nine-dimensional output vector, representing the model's predicted probabilities for the nine distinct categories.

### B. The Multi-Branch MACRO (MB-M) Architecture

Inspired by Zhang et al.'s MLBF-Net [30], we extend the MACRO architecture to create Multi-Branch MACRO (MB-M). This enhancement includes twelve lead-specific branches, each tailored to the unique characteristics of different ECG leads. These branches are then consolidated into a concatenated overall network, providing a holistic view that considers all twelve ECG channels for signal classification. Fig. 3 shows an overview of the proposed MB-M model and the main components are described in the following paragraphs.

*1) Twelve Independent BranchNets:* Each BranchNet closely resembles the MACRO architecture (cf. Fig. 1), with minor alterations in the number of channels, processing a single lead's ECG signal instead of the full 12-lead ECG. Hence, all hyperparameters remain consistent with MACRO, except for the number of channels in the input and consequently, between the five convolutional block, as indicated at the bottom of Fig. 3.

*2) Branch Fusion and Convolutional Reduction Block:* This module concatenates all feature maps $h_t^{BiRNN} \in \mathbb{R}^{24}$ from the BiGRUs across all branches along the channel axis. Subsequently, it gradually decreases the concatenated channels

from $24 * 12 = 288$ to 24. This ensures that the dimensions of the subsequent MHA layer matrices remain within a reasonable range. The reduction is accomplished through three convolutional layers, each, except the last, followed by batch normalization and LeakyReLU activation layers. Notably, the final convolution lacks the normalization, which aligns with the original MACRO architecture, where the BiGRU feature maps remain unnormalized before entering the MHA layer.

*3) MHA Layer and Classification Head:* The last two components maintain the structure of MACRO. Firstly, the concatenated and reduced BiGRU feature maps undergo a MHA mechanism with $h = 24$ heads. Secondly, they pass through the classification head, which is identical to MACRO.

*4) Multi-Loss Objective Function:* To jointly optimize the concatenated network and the twelve individual branches, we employ a multi-loss objective function structured as follows:

$$L = L_c + \lambda(L_1 + ... + L_{12}) \qquad (5)$$

Here, $L_c$ and $L_i$ represent the binary cross-entropy (BCE) loss of the concatenated network and the $i$-th BranchNet, respectively. Following insights from Zhang et al. [30], we set the parameter $\lambda$ to 1 to effectively balance the contributions of lead-specific features from individual branches and the comprehensive features derived from the overall network.

### C. MB-M and Gradient Boosting Classifiers

Previously, individual BranchNets were optimized independently with a multi-loss approach, but their direct contribution to final classification was absent. In a shift towards ensemble modeling principles, we repurpose the deep learning architecture MB-M as a feature extractor, and introduce gradient boosting models for classification [34]. Employing a dedicated binary classifier for each of the nine classes, we experiment with three variants of input features:

1) Predicted probabilities (PPs) from all twelve BranchNets and MB-M (117 feat.)
2) PPs only for the class of interest (13 feat.)
3) PPs only for the class of interest w/o MB-M (12 feat.)

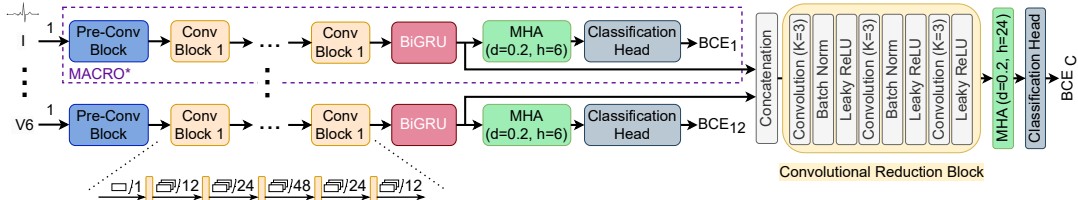

Fig. 3: Overview of the proposed Multi-Branch MACRO architecture

## III. EXPERIMENTAL SETUP

### A. Datasets

Our architectures are developed and assessed on the widely used benchmark dataset of the China Physiological Signal Challenge (CPSC) 2018 [35]. This real-world dataset from 11 hospitals encompasses nine classes, including sinus normal rhythm (SNR) and eight rhythmical and morphological abnormalities. We utilize the 6,877 records from the publicly available training set, as the test set remains private. Most records are 6 to 60 seconds long, with a few extending up to 144 seconds. While primarily single-labeled, 476 records have multiple labels. To assess generalizability, we utilize PTB-XL [36], a large multi-label dataset of 21,799 clinical 12-lead ECG records of 10 seconds each. PTB-XL contains 71 ECG statements, categorized into 44 diagnostic, 19 form, and 12 rhythmic classes. In addition, the diagnostic category can be divided into 24 sub- and 5 coarse-grained super-classes. In alignment with the recent SOTA method proposed by Tao et al. [37], we utilize the super-diagnostic labels for classification.

### B. Data Management and Preprocessing

In initial experiments and for hyperparameter tuning, we employed a fixed split of the 6,877 samples from the CPSC 2018 dataset: 60% for training, 20% for validation, and 20% for testing. For the final evaluation, including ablation studies and model comparisons, we used 10-fold cross-validation (CV) with random partitions. Our preprocessing, tailored for clinical practicality, remains minimal: signals are downsampled from 500 Hz to 250 Hz, and record durations are standardized to 60 seconds. Over-length samples are truncated, and shorter ones are zero-padded. For PTB-XL, we used the recommended train-valid-test splits, sampled at 100 Hz to ensure comparability with existing methods. Moreover, we select only samples with at least one label in the superdiagnostic category, without applying any further preprocessing.

### C. Evaluation Metrics

We assess model performance using established metrics, such as $F_1$ score, AUC, and subset accuracy. In binary classification, TP, TN, FP, and FN represent true positives, true negatives, false positives, and false negatives, respectively. Accuracy (Acc), Precision (Prec), Recall (Rec), and $F_1$ score are defined as:

$$Acc = \frac{TP + TN}{TP + FP} \quad (6) \qquad Recall = \frac{TP}{TP + FN} \quad (8)$$

$$Prec = \frac{TP}{TP + FP} \quad (7) \qquad F_1 = \frac{2 \times Prec \times Rec}{Prec + Rec} \quad (9)$$

AUC quantifies the area under the curve generated by plotting the true positive rate (TPR) against the false positive rate (FPR) at different discrimination thresholds. For multi-label classification, we apply weighted and macro averaging. Weighted averaging considers class contributions based on support, addressing class imbalance, while macro averaging is a common practice in recent approaches. The exact match ratio (MR) is also used to assess the percentage of samples with all labels correctly predicted.

### D. Implementation Details

We implemented our approach using Python 3.10.12 and PyTorch 2.0.1 on a server with two partitioned Nvidia A100 GPUs, each with 80 GB RAM. The baseline, MACRO, and MB-M models were trained end-to-end using the Adam optimizer with default settings, a learning rate of 0.001, and a batch size of 64. We used binary cross-entropy (BCE) loss and applied early stopping with a patience of 20 epochs on the validation set. We saved the model's weights at the time of the last improvement and opted for the macro $F_1$ score as stopping criterion. To train and fine-tune the gradient boosting (GB) classifiers for each class of CPSC 2018, we reemployed the 10-fold cross-validation (CV) approach with data splits identical to those used in the end-to-end MB-M training to prevent data leakage and maintain test set integrity. In each iteration, we merged the folds of the training and validation sets to form a new training set. Using this set, *sklearn*'s stratified k-fold CV (k=3) and grid search were used to fine-tune GB hyperparameters for each class. The best per-class models were then applied to the probabilities generated by the BranchNets and MB-M for the corresponding test fold, depending on the feature selection. The final classification resulted from combining the binary predictions of each individual GB classifier. For PTB-XL, we followed a similar process but used the recommended train-valid-test split instead of cross-validation.

## IV. RESULTS

### A. Classification Performance of Our Models

The 10-fold CV results for our MACRO and Multi-Branch MACRO (MB-M) models, compared to our reimplemented baseline [10], are given at the left of Table I. The $F_1$ score and AUC metrics, reflecting the models' predictions on nine distinct classes, are averaged across respective test datasets from the ten rounds. These average values, along with the standard deviation (sd), are reported. Additionally, macro and weighted averages across all classes and folds are provided. The input for all three models remains consistent ($x \in \mathbb{R}^{12 \times 15,000}$).

TABLE I: 10-fold CV results (mean±sd) in percentage on the CPSC 2018 dataset. N denotes the number of samples per class.

| Type | (N) | Baseline [10] | | MACRO | | MB-M | | MB-M + GB-12 | | MB-M + GB-13 | | MB-M + GB-all | |
|---|---|---|---|---|---|---|---|---|---|---|---|---|---|
| | | $F_1$ | AUC | $F_1$ | AUC | $F_1$ | AUC | $F_1$ | AUC | $F_1$ | AUC | $F_1$ | AUC |
| SNR | (918) | 79.6±3.7 | 97.2±0.9 | 81.5±3.0 | 97.5±0.5 | **82.8±1.6** | **97.8±0.3** | 79.3±3.0 | 97.7±0.4 | 83.0±1.9 | 97.8±0.5 | **83.1±2.6** | **97.9±0.4** |
| AF | (1221) | 90.3±1.8 | 98.6±0.5 | 92.1±1.4 | 98.9±0.5 | **93.7±1.4** | **99.1±0.4** | 93.9±1.6 | 99.2±0.4 | 94.1±1.3 | 99.2±0.3 | **94.6±1.1** | 99.2±0.5 |
| IAVB | (722) | 86.1±2.7 | 98.5±0.9 | 86.8±4.0 | 98.2±1.1 | **88.6±3.6** | **99.0±0.5** | **89.2±3.3** | **99.1±0.5** | **89.2±3.6** | 98.9±0.7 | 88.8±3.6 | 99.0±0.6 |
| LBBB | (236) | 85.0±8.9 | 98.1±2.0 | 86.3±7.5 | **98.6±1.3** | 86.6±5.7 | 98.4±2.2 | 89.0±5.7 | 98.7±1.6 | 89.6±5.1 | **98.8±1.4** | **90.3±6.4** | 98.3±2.1 |
| RBBB | (1857) | 92.0±1.3 | 98.6±0.4 | 92.7±1.1 | **98.9±0.3** | 92.9±0.8 | **98.9±0.2** | **94.1±0.8** | **99.1±0.3** | 93.8±1.0 | 99.0±0.3 | 93.9±0.8 | **99.1±0.3** |
| PAC | (616) | 72.2±2.5 | 95.3±1.3 | 73.4±4.8 | 96.4±1.6 | **79.2±4.0** | 97.4±1.3 | 80.2±4.9 | **98.2±0.9** | 79.1±4.7 | 98.1±0.9 | **81.4±3.8** | 97.7±1.4 |
| PVC | (700) | 84.7±4.5 | 98.0±0.7 | 85.4±2.3 | **98.3±0.6** | 86.0±2.1 | 97.9±0.8 | **88.1±1.7** | **98.8±0.4** | 87.8±2.0 | 98.5±0.6 | 88.0±1.9 | 98.4±0.9 |
| STD | (869) | 77.0±4.0 | 94.8±2.0 | **79.1±4.0** | 96.1±1.4 | 79.0±3.4 | **96.3±1.1** | 82.6±3.2 | **97.4±0.7** | 83.2±3.4 | 97.2±0.8 | **83.4±2.6** | **97.4±0.7** |
| STE | (220) | 43.1±12.0 | 90.0±4.7 | 51.0±11.1 | 92.4±2.8 | **55.7±6.2** | **94.4±3.8** | 60.3±6.0 | 94.4±2.7 | **64.3±9.0** | **95.7±2.8** | 63.0±5.1 | 95.4±2.5 |
| Macro-AVG | | 78.9±1.6 | 96.6±0.6 | 80.9±2.1 | 97.3±0.5 | **82.7±1.4** | **97.7±0.6** | 84.1±1.1 | **98.1±0.4** | 84.9±1.2 | **98.1±0.3** | **85.2±1.2** | 98.0±0.5 |
| Weighted-AVG | | 83.9±1.2 | 97.4±0.4 | 85.4±1.3 | 97.8±0.4 | **86.7±1.1** | **98.1±0.3** | 87.5±1.2 | **98.5±0.2** | 88.1±1.0 | 98.4±0.2 | **88.4±0.8** | 98.4±0.3 |

In summary, the transition from the baseline to MACRO and to MB-M results in a noticeable improvement across multiple metrics. Specifically, there is an increase of 2.0 pp and 3.8 pp in macro-averaged $F_1$ and 0.7 pp and 1.1 pp in AUC scores, respectively. For weighted averages, the increments are 1.5 pp and 2.8 pp for the $F_1$ and 0.4 pp and 0.7 pp for the AUC scores. When examining the $F_1$ scores for each class, there is a consistent increase in mean performance scores across all ten folds for all classes. The only exception is a negligible decrease of 0.1 pp for STD when comparing MACRO to MB-M. Similar observations can be made for the class-specific AUC scores, with only three minor exceptions. Furthermore, all three models consistently demonstrated proficiency in the two majority classes, RBBB and AF, but faced challenges in effectively handling the minority class STE. Interestingly, the second minority class, LBBB, was quite well recognised, with all three models achieving an average $F_1$ score of at least 0.85. Although the sample sizes for STE (220) and LBBB (236) were similar, this difference in performance may be due to the considerable disagreement between physicians in diagnosing STE from ECGs [38]. The complexities in recognizing STE are further underscored by examining the standard deviation (sd) across the 10 folds. For STE, all three models showed the highest sd in $F_1$ scores, followed by the other minority class LBBB, which had the second-highest sd across the models.

As shown at the right-hand side of Table I, the usage of gradient boosting (GB), utilizing diverse input features, considerably improves the overall performance. For all three variants, there is a consistent improvement over MB-M of 1.4 pp, 2.2 pp, and 2.5 pp in the macro average $F_1$ scores, accompanied by a macro AUC score increase of $\geq 0.3$ pp. In particular, the minority classes STE and LBBB experience substantial improvement in all cases. When using GB classifiers with all 117 input features, we observe enhancements in all class-wise $F_1$ scores and both averages. The AUC scores improve or remain constant for all labels, except for LBBB. Restricting the GB input features to the BranchNets and MB-M outputs for the specific class of interest (13 ft.) yields similar positive outcomes, with increased $F_1$ scores for all classes except PAC and improved or consistent AUC scores, except for I-AVB. Further limiting the GB inputs by excluding the MB-M probability (12 ft.) follows a similar pattern of improved $F_1$ scores and either enhanced or unchanged AUC scores compared to the raw MB-M. Notably, the sole exception is observed in normal sinus rhythm (SNR), where there is a notable 3.5pp decrease in the $F_1$ score. To provide context, the few exceptions noted for the other GB variants yield a decrease of only 0.01 pp.

While the integration of GB classifiers contributes significantly to improved performance regardless of the specific feature set, no consistent trends are observed within the different GB variants. A stringent rise in $F_1$ scores, moving from 12 to 13 to all 117 features, is evident for only four of the nine classes (SNR, AF, LBBB, and STD), along with macro and weighted averages. In the case of I-AVB, RBBB, and PVC, both $F_1$ and AUC scores show minor differences ($\leq 0.5$ pp) across feature sets. Conversely, for PAC and STE, more substantial variations are observed across both metrics. Consequently, our focus will be on two versions: GB-all, utilizing all 117 features and demonstrating the best overall performance, and GB-13, incorporating only 13 features, while exhibiting superior performance for the minority class STE.

### B. Comparison to Existing SOTA Techniques

This section compares the performance of our method with previous work for 12-lead ECG classification on the CPSC 2018 dataset, provided that the respective methods have been evaluated by 5- or 10-fold CV. As shown in Table II, MACRO and in particular, Multi-Branch MACRO (MB-M) without gradient boosting (GB) achieve competitive results compared to recent state-of-the-art (SOTA) methods. When MB-M is used as a feature extractor combined with GB classifiers, it outperforms existing SOTA approaches, demonstrating superior $F_1$ scores across various classes and, especially, the macro $F_1$, AUC, and Acc scores. In more detail, the combination of MB-M and the GB-all classifier surpasses current SOTA methodologies in terms of all three macro averages and three out of nine class-specific $F_1$ scores. Even when using only 13 input features per classifier (GB-13), our approach achieves superior performance. This includes improved macro $F_1$, AUC, and Acc scores, as well as a superior F1 score for class I-AVB compared to existing techniques. Thus, the proposed architecture effectively addresses the complexities of detecting concurrent cardiac disorders.

TABLE II: Comparison of our 10-fold CV results (in %) with exist. SOTA methods applying 5- or 10-fold CV on CPSC 2018.

| Approach | | | Class-wise $F_1$ | | | | | | | | | Macro AVG | | |
|---|---|---|---|---|---|---|---|---|---|---|---|---|---|---|
| Method | Year | SNR | AF | I-AVB | LBBB | RBBB | PAC | PVC | STD | STE | $F_1 \downarrow$ | AUC | Acc |
| Res. att. modules + LSTM [26] | 2019 | 80.0 | 84.5 | 83.3 | 81.0 | 87.2 | 73.1 | 81.8 | 79.0 | 55.3 | 78.4 | - | - |
| ResNet + BiLSTM [24] | 2019 | 75.5 | 84.6 | 87.0 | 86.9 | 78.0 | 75.1 | 82.9 | 79.1 | **70.4** | 79.9 | - | - |
| LightX3ECG [21] | 2023 | 75.5 | 94.0 | **89.2** | 88.7 | **94.4** | 63.1 | 79.2 | 78.5 | 57.8 | 80.0 | - | - |
| CNN + BiLSTM [15] | 2020 | 79.9 | 91.7 | 88.1 | 88.1 | 93.9 | 59.3 | 81.2 | 81.4 | 58.8 | 80.3 | 96.2 | 96.2 |
| ResNet + expert features [23] | 2018 | 82 | 91 | 87 | 87 | 91 | 63 | 82 | 81 | 60 | 81 | - | - |
| (Interpretable) ResNet [18] | 2021 | 80.5 | 91.9 | 86.4 | 86.6 | 92.6 | 73.5 | 85.1 | 81.4 | 53.5 | 81.3 | 97.0 | 96.6 |
| ASTLNet [20] | 2023 | 79.0 | 92.3 | 86.7 | 89.2 | 93.7 | 75.7 | 83.7 | 79.8 | 55.8 | 81.8 | 97.0 | 80.0 |
| ResNet with SE blocks [25] | 2021 | 79 | 92 | 87 | 87 | 93 | 78 | 86 | 81 | 59 | 82.5 | - | - |
| Multi-task neural network [27] | 2023 | 82.4 | 92.5 | 88.2 | **93.7** | 93.9 | 73.4 | 76.7 | 83.5 | 60.0 | 82.7 | 97.7 | 96.6 |
| DAMS-Net [28] | 2023 | 81.9 | 91.5 | 88.1 | 87.8 | 93.6 | 75.5 | 87.6 | 81.9 | 68.4 | 83.9 | - | - |
| LFG-Net [29] | 2024 | 79.2 | 93.2 | 89.1 | 89.4 | 93.7 | 75.6 | 87.4 | 82.1 | 68.2 | 84.2 | - | - |
| MSGformer [22] | 2024 | **84.0** | 92.3 | 83.8 | 84.9 | 93.5 | 73.1 | 85.6 | **85.6** | 59.8 | 84.7 | - | - |
| MACRO | 2024 | 81.5 | 92.1 | 86.8 | 86.3 | 92.7 | 73.4 | 85.4 | 79.1 | 51.0 | 80.9 | 97.3 | 96.6 |
| MB-M | 2024 | 82.8 | 93.7 | 88.6 | 86.6 | 92.9 | 79.2 | 86.0 | 79.0 | 55.7 | 82.7 | 97.7 | 96.9 |
| MB-M + GB-13 | 2024 | 83.0 | 94.1 | **89.2** | 89.6 | 93.8 | 79.1 | 87.8 | 83.2 | 64.3 | 84.9 | **98.1** | **97.2** |
| MB-M + GB-all | 2024 | 83.1 | **94.6** | 88.8 | 90.3 | 93.9 | **81.4** | **88.0** | 83.4 | 63.0 | **85.2** | 98.0 | **97.2** |

## C. Ablation Studies and Amount of Parameters

We evaluate different model variants by 10-fold CV to understand their contributions. Our baseline, adapted from Chen et al. [10], is compared to MACRO, its multi-branch version and two intermediate alternatives in terms of macro (m) and weighted (w) $F_1$ and AUC scores, as well as MR. To examine the impact of the multi-head attention (MHA), we replace the simple attention in the baseline while optimizing MHA parameters for the simplified CNN. Identified optimal parameters ($h = 8$, *dropout* $= 0.4$) yield marginal improvements with a negligible parameter increase. Conversely, substituting the CNN module enhances performance across metrics, notably a 1.7pp increase in macro $F_1$, but comes with a significant rise in trainable parameters. Since the CNN module is the first component of the network and does not require any adaptation to preceding layers, it is replaced with MACRO's version without parameter tuning. The combined variant, featuring both the improved CNN and the MHA module (resulting in MACRO), exhibits further improvements across metrics. Although marginal compared to the baseline with modified CNN, these enhancements are noteworthy compared to both the baseline and its MHA-only version. Ultimately, the fusion of twelve MACRO models into MB-M leads to additional performance gains across all metrics, albeit with a further increase in trainable parameters. Nevertheless, MB-M maintains a low number of trainable parameters compared to existing SOTA approaches. Table IV summarizes this for all methods from Table II that provide either information on the number of parameters or their source code. With the exception of LFG-Net [29], which requires 1.02M parameters for pure inference, but necessitates the use of another 12 parallel networks for training, and He et al.'s method [24], whose achieved macro $F_1$ score (m-$F_1$) is 5 pp. below ours, our approach achieves a reduction in parameters of between 29% (wrt. [20]) and 68% (wrt. [21]). This underscores a thoughtful balance between model complexity and computational efficiency on the one hand, and classification performance on the other hand.

TABLE III: Performance of model variants by 10-fold CV.

| Approach | m-$F_1$ | w-$F_1$ | m-AUC | w-AUC | MR | Params $\downarrow$ |
|---|---|---|---|---|---|---|
| Baseline (BL) | 78.9 | 83.9 | 96.6 | 97.4 | 78.6 | 28,005 |
| BL + MHA | 79.3 | 84.1 | 96.8 | 97.4 | 78.7 | 29,805 |
| BL + mod. CNN | 80.6 | 85.1 | 97.2 | 97.8 | 79.9 | 190,113 |
| MACRO | 80.9 | 85.4 | 97.3 | 97.8 | 80.0 | 191,913 |
| MB-M | **82.7** | **86.7** | **97.7** | **98.1** | **81.9** | 1,713,081 |

TABLE IV: Parameter count $P_C$ in million of MACRO (M), MB-M, and MB-M + GB-all (ours) compared to others.

| | [24] | [21] | [18] | [20] | [25] | [29] | M | MB-M | Ours |
|---|---|---|---|---|---|---|---|---|---|
| $P_C$ | 1.16 | 5.34 | 3.87 | 2.42 | 3.5 | 1.02 | **0.19** | 1.71 | 1.71 |
| m-$F_1$ | 79.9 | 80.0 | 81.3 | 81.8 | 82.5 | 84.2 | 80.9 | 82.7 | **85.2** |

## D. Investigation of Generalizability

To assess the generalizability of our approach, we trained and evaluated MACRO, MB-M, and its GB-all extension without modifications on the super-diagnostic task of PTB-XL. Table V presents and compares our results with other methods.

TABLE V: Results and comparison with others on PTB-XL.

| Method | Year | $P_C$ | m-$F_1 \downarrow$ | m-AUC | m-Acc |
|---|---|---|---|---|---|
| LightX3ECG [21] [1,2] | 2023 | 5.34 | 71.9 | 92.0 | 88.4 |
| ECG-DNN [19] [1,2] | 2019 | 10.48 | 71.9 | 92.4 | 88.4 |
| SE-ResNet12 [1,3] | 2022 | ≈45 | 72.5 | 92.3 | 88.0 |
| Resnet34_1d [1,3] | 2016 | ≈7.2 | 72.6 | 90.8 | 88.2 |
| ASTLNet [20] [2] | 2023 | 2.42 | 73.6 | 91.3 | 62.7 |
| Xresnet1d101 [1] | 2021 | 1.81 | 73.7 | **92.9** | 88.5 |
| Image_CNN [1,2] | 2018 | 135.38 | 74.1 | 92.1 | 88.8 |
| DNN_zhu [1] | 2020 | n/a | 76.2 | 91.8 | 89.0 |
| 2D-ECGNet [37] [1] | 2024 | n/a | **77.0** | **92.9** | **89.2** |
| MACRO | | **0.19** | 73.1 | 92.0 | 87.6 |
| MB-M | | 1.71 | 74.9 | 92.4 | 88.8 |
| Ours (MB-M + GB-all) | | 1.71 | 74.6 | 92.7 | 89.1 |

[1] Models detailed in [37]   [2] $P_C$ (in M) estimated from existing code
[3] $P_C$ roughly estimated from similar model code (SE-ResNet152/ResNet34)

While the transition from MACRO to MB-M results in improvements of 1.8 pp in $F_1$, 0.4 pp in AUC, and 1.1 pp in accuracy on PTB-XL, these gains are less pronounced than those observed on CPSC. The addition of GB-all classifiers provides only minimal benefits of 0.3 pp regarding AUC and accuracy. This limited impact may stem from PTB-XL's minority classes: With the lowest class representing around 10% of samples, they may not experience the same boost from binary classifiers as seen on CPSC, where the STE class constitutes just 3%. In comparison to existing SOTA techniques, our approach achieves comparable results while simultaneously exhibiting a relatively low parameter count.

## V. CONCLUSION

In this study, we proposed the MACRO and Multi-Branch MACRO (MB-M) architectures for detecting concurrent cardiac abnormalities in 12-lead ECG signals. Both models integrate CNNs and RNNs with a multi-head attention mechanism, achieving highly competitve performance on the CPSC 2018 dataset and demonstrating good generalizability, as indicated by competitive results on the PTB-XL dataset. While MACRO processes the 12-lead ECG input as a whole, the MB-M model integrates twelve lead-specific branches into a comprehensive architecture to provide a holistic understanding of cardiac activity. It adeptly captures the unique characteristics of different ECG leads, improving the classification of different abnormalities beyond the already strong capabilities of MACRO. By repurposing MB-M as a feature extractor to train individual gradient boosting (GB) classifiers for each label, we further increased classification performance on CPSC 2018, particularly for minority classes. The combination of MB-M and GB classifiers outperforms current SOTA methods in 10-fold cross-validation on CPSC 2018, while reducing trainable parameter count by up to 68%. In addition to its parameter efficiency and SOTA performance on a widespread ECG dataset, our approach has been shown to be effective across different data splits, as demonstrated in extensive evaluations and ablation studies. Lastly, we advance the field of cardiac abnormality detection by providing insights into ensemble modelling using a novel combination of deep feature extractors and traditional machine learning classifiers. In the future, we plan to evaluate the generalizabilty of our method in more depth through further experiments, including other PTB-XL categories, and to assess its transfer learning potential.

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
