# MACRO: Supplementary Material

Vanessa Borst (iD) (✉), Robert Leppich (iD) and Samuel Kounev (iD)

University of Würzburg, Würzburg, Germany

Email: {vanessa.borst, robert.leppich, samuel.kounev}@uni-wuerzburg.de

## I. RELATED WORK

In this section, the history of automatic detection of cardiovascular diseases (CVD) based on electrocardiograms (ECG), which was outlined within the introduction of the main paper, is supplemented by more details and a comprehensive list of examples. First, an overview of traditional machine learning techniques is presented in Subsection I-A, followed by a summary of existing deep learning techniques in Subsection I-B. Finally, advanced deep learning mechanisms that have been deployed for CVD detection are reviewed in Subsection I-C.

### A. Traditional Machine Learning Techniques

Traditional machine learning for ECG analysis involves multiple steps. Initially, the raw ECG signal is pre-processed to eliminate noise, baseline wandering, and other artifacts, optionally also segmenting heartbeats. Then, feature extraction is performed, considering various signal properties like the the signal's higher order statistics or its morphology in the time/frequency domain. Techniques like independent component analysis, autoregressive modeling, Hermite basis functions, and especially, different variants of wavelet transforms (e.g., DWT, CWT, or XWT) are commonly used. Dimensionality reduction methods, such as PCA or LDA, and feature ranking procedures may follow feature extraction. Finally, for CVD detection, a range of supervised and unsupervised methods is available, including SVMs, Bayesian classifiers, random forests, k-nearest neighbor (KNN) classifiers, and multi-layer perceptrons with varying architectures. Unsupervised approaches, such as heuristic-search-based clustering and two-dimensional Gaussian spectral clustering, have also been explored.

### B. Deep Learning Methods

Deep learning (DL) has revolutionized ECG analysis, eliminating the need for extensive preprocessing and manual feature extraction. Recent methods often perform end-to-end processing, combining feature extraction and classification [18]–[20], with some using raw ECG data directly [21] or with minimal modifications like padding [22]. An overview of existing methods is provided in Table II.

Convolutional Neural Networks (CNNs) have been extensively explored for ECG classification at both the heartbeat level and in methods operating on ECG sequences. While CNNs excel at extracting discriminative features from spatially and locally related data, they often neglect the temporal properties of ECG signals [23]. In contrast, Recurrent Neural Networks (RNNs), such as LSTMs and GRUs, are designed

TABLE I: Overview of traditional machine learning techniques for automated CVD detection, including feature extraction techniques

| | |
|---|---|
| Feature Extraction | • Independent component analysis [1]
• Autoregressive modeling [2]
• Hermite basis functions [3]
• Variants of wavelet transforms (e.g., DWT [4], CWT [5], XWT [6], or FAWT [7]) |
| Dimensionality Reduction | • Principal component analysis(PCA) [4]
• Linear discriminant analysis (LDA) [8]
• Locality perserving projection (LPP) [9]) |
| Feature Revision | • Feature ranking [7], [9], [10]
• Feature normalization [11] |
| Classification | • SVM [4], [7]
• Bayesian classifier [12]
• Random Forest (RF) [13]
• k-nearest neighbor (KNN) classifier [10], [14]
• Multi-layer perceptrons (MLP) [4], [15]
• Unsupervised methods (e.g., heuristic-search-based clustering [16], or two-dimensional Gaussian spectral clustering [17]) |

for handling sequential data of varying lengths but lack spatial information [23]. Capturing the intricate temporal dynamics inherent in ECGs, they have been applied for both the classification on heartbeat-level and on the sequence-level as well. Hybrid architectures, leveraging the strengths of both CNNs and RNNs, have also proven successful in CVD detection. While some researchers stack both network types in arbitrary order, others apply them in parallel. Furthermore, there are methods that first train an autoencoder (AE) with unsupervised learning and later use its compressed representation for subsequent classification, including convolutional AEs, LSTM-based AEs, and stacked denoising AEs.

TABLE II: Overview of previous deep learning techniques for CVD detection

| | |
|---|---|
| Convolutional Neural Network (CNN) | • Heartbeat level [24], [25]
• Sequence level [20], [26] |
| Recurrent Neural Network (RNN) | • Heartbeat level [27], [28]
• Sequence level [29] |
| Hybrid (CNN + RNN)* | • Stacked - CNN first [18], [21], [30], [31]
• Stacked - RNN first [32]
• In parallel [33] |
| Autoencoder (AE)* | • Convolutional (possibly denoising) AE [34], [35]
• LSTM-based AE [36]
• Stacked, denoising AEs [37] |

* No distinction is made between hearbeat- and sequence level

## C. Advanced Deep Learning Concepts

Within the last decade and especially, within the last five years, many architectures incorporating advanced deep learning concepts, such as residual networks or attention mechanisms, have been proposed. As indicated in the (non-exhaustive) summary of Table III, advanced methods for 12-lead CVD detection, mostly from the last five years, include CNNs with dilated and deformable convolutions, residual neural networks, attention mechanisms, and transformer-based architectures. Residual networks (ResNets) exhibit numerous variants, such as the integration of handcrafted (expert) features or their combination with RNNs or transfer learning. ResNets with attention modules, such as squeeze-and-excitation (SE) blocks or convolutional block attention (CBAM) modules, and methods that merge (SE)ResNets with transformer variants have also been successfully applied. Lastly, approaches that process different ECG leads separately or in groups by dedicated feature extraction components per lead (group) within different network branches have made notable contributions.

TABLE III: Overview of advanced DL for CVD detection

| Advanced Convs. | • Dilated convolution [38]–[40]
• Deformable convolution  [41] |
|---|---|
| Residual Networks | • Simple ResNets [42]–[44]
• ResNet + hand-crafted expert features [45]
• ResNet + RNNs [46]
• ResNet + transfer learning [47]
• ResNet + Attention blocks
  (e.g., SE blocks [48]–[50],
  or CBAM modules [51]–[53]) |
| RNNs with Attention | • LSTM [54]–[56]
• GRU [22], [57] |
| Transformer-based | • Transformer encoder [58]–[60]
• Transformer as a whole [61] |
| Others | • (SE)ResNets + Transformer variants [62], [63]
• Separate lead processing by dedicated
  network components (e.g., single lead
  processing [57], [64], [65], or grouped
  lead processing [66]) |

## II. METHODOLOGY

This section begins with the problem formulation in Section II-A. Then the challenge-best model of Chen et al. [22] is explained in Section II-B, including a visualization of the employed attention mechanism. Finally, Section II-C provides more details about our MACRO architecture.

### A. Problem Formulation

The detection of cardiac irregularities within ECG signals can be framed as a time-series classification task. In a 12-lead multi-label classification context, the model operates on ECG recordings of variable lengths, denoted as $x_i \in \mathbb{R}^{L \times 12}$, where $L$ represents the length of a given recording $x_i$ (in our work $L = 15,000$). The aim of the model is to predict one or more classes from a pool of $C$ potential classes for each individual recording, concurrently. Hence, the objective function of the model seeks to minimize the binary cross entropy loss $L_{BCE}$ between the actual ground truth labels associated with a given recording $x_i$ and the labels predicted by the model for the same record, which is defined as follows:

$$L_{BCE}(\hat{Y}_i, Y_i) = -\sum_{j=1}^{C} y_j log(\hat{y}_j) + (1 - y_j)log(1 - \hat{y}_j) \quad (1)$$

where $Y_i = (y_1, ..., y_C)$ with $y_i \in \{0, 1\}$ represents the ground truth label annotations for sample $x_i$ and $\hat{Y}_i = (\hat{y}_1, ..., \hat{y}_C)$ with $0 \leq \hat{y}_i \leq 1$ denotes the vector containing the class probabilities predicted for $x_i$ by the multi-label classifier.

### B. Challenge-Best Model of Chen et al. [22]

*1) Baseline Architecture:* The baseline model of Chen et al. [22] comprises distinct neural network elements, grouped into a CNN component, an RNN module, an attention mechanism, a normalization layer, and a final fully connected unit. Fig. 1 illustrates the overall architecture, with the CNN segment consisting of five nearly identical CNN blocks. Operating on raw 12-lead ECG values, the number of incoming and outgoing channels stays the same for all five CNN blocks and amounts to twelve. Each block employs two 1D convolutional layers with a kernel size of three and stride one, followed by a downsampling layer to reduce complexity and over-fitting. Notably, in the first four CNN blocks, the downsampling is realized as convolution with stride two and kernel size 24, while in the last block, a 48-sized kernel is used.

The outputs of the last CNN block, per time step consisting of twelve features, are processed by a bidirectional gated recurrent unit (BiGRU) with one layer and twelve units, which generates a 24-dimensional vector per time step by concatenating its forward and backward outputs. The subsequent attention layer determines importance weights for the BiGRU's hidden states at different time steps, yielding a weighted sum across all input features as a 24-dimensional output.

Batch normalization precedes the final dense layer that maps its inputs to a nine-dimensional output vector of classification probabilities. LeakyReLU with a negative slope of 0.3 is employed as the activation function, except for the dense layer, which uses a Sigmoid activation. Dropout randomly omits 20% of connections between CNN blocks and between other independent layers to enhance robustness.

*2) Attention Mechanism:* For determining the importance weight of a given hidden state $h_t^{BiRNN}$, the concatenated vector is first transformed to a hidden representation $u_t \in \mathbb{R}^{24}$ by a one-layer MLP, with $W \in \mathbb{R}^{24 \times 24}$ and $b \in \mathbb{R}^{24}$ denoting learnable parameters and $tanh$ serving as activation:

$$u_t = tanh\left(Wh_t^{BiRNN} + b\right) \quad (2)$$

Afterwards, the importance of the respective hidden state is calculated as the similarity between $u_t$ and a query vector $q \in \mathbb{R}^{24}$, where the dot-product is used as attention scoring function. The query vector $q$ is randomly initialized and jointly learned during the training procedure and hence, it can technically be realized as a one-layer MLP with a weight matrix of shape $24 \times 1$ and no bias.

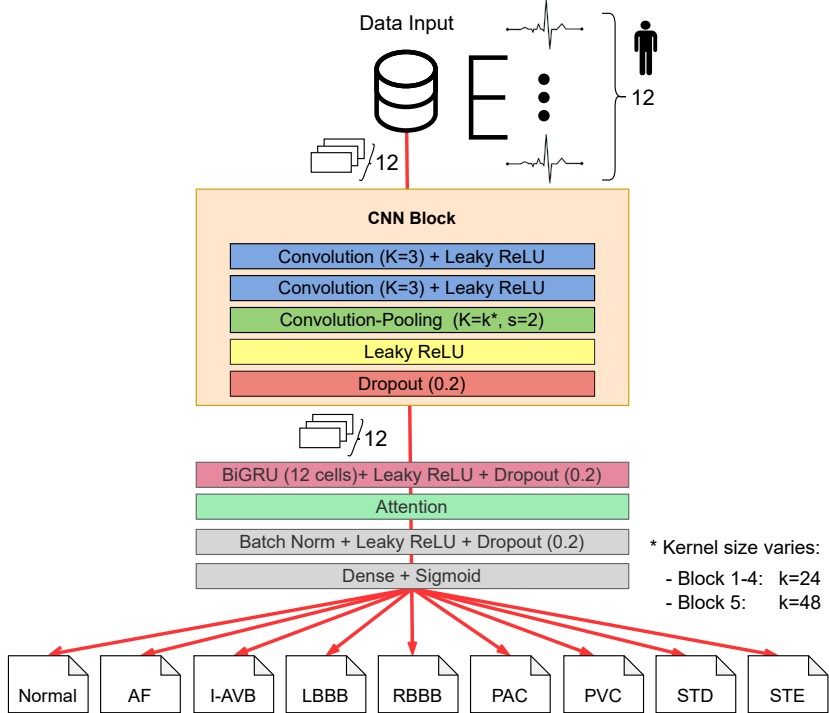

Fig. 1: Model architecture used in the ensemble approach of Chen at al [22]. Our reimplementation of this approach serves as a baseline in this work.

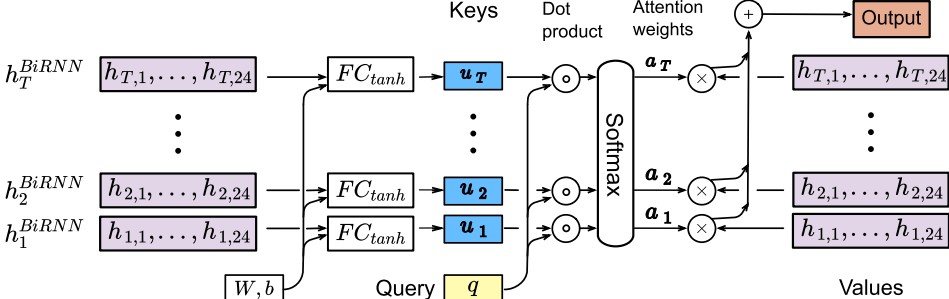

Fig. 2: Application of the attention framework to the hidden states of a BiRNN with 12 units. This attention mechanism is used in our reimplementation of the approach of Chen et al. [22].

Thereupon, the computed, scalar attention score is further processed by a Softmax function to retrieve a normalized importance weight $\alpha_t$ for time step $t$:

$$\alpha_t = Softmax\left(q^T u_t\right) = \frac{exp(q^T u_t)}{\sum_{j=1}^{T} exp(q^T u_j)} \qquad (3)$$

At the end, the output of the attention layer is computed as weighted sum over the $T$ time steps as follows, yielding a final vector $f_{att} \in \mathbb{R}^{24}$:

$$f_{att} = \sum_{t=1}^{T} \alpha_t h_t^{BiRNN} \qquad (4)$$

An illustration of this mechanism can be found in Fig. 2.

## C. MACRO: Design Choices and Multi-Head Attention

*1) Design Rationale Behind the CNN Module:* This subsection provides more details on our design rationale regarding the CNN module. Based on preliminary experiments, we have found that the following mechanisms are beneficial to the final classification performance compared to the original CNN submodule of Chen et al. [22]:

1) **Skip Connections**: We introduce skip connections within all network blocks to facilitate the direct flow of gradients throughout the neural network. We organize these skip connections following the pre-activation design paradigm, inspired by a seminal study by He et al. [67]. He et al. conducted a comprehensive analysis of residual blocks shortly after introducing skip connections and ResNets in their pioneering work [68]. They explored various configurations of skip connections and argued that establishing a "clean" pathway for direct information propagation, not only within a residual unit but across the entire network, is advantageous for optimization and generalization. Notably, in the pre-activation design, the shortcut path omits normalization and activation functions to closely approximate an identity mapping by immediately capturing the resulting signal from the addition operation and transmitting it to the subsequent residual block, even prior to normalization. Consequently, information passed through the skip connection propagates almost directly between different units, except for the channel and spatial alignment operation.

2) **Up-Front Block**: Because of the pre-activation design, a block is added to the beginning of MACRO before the five convolutional blocks, ensuring that the ECG signal, which is only down-sampled and length-restricted but otherwise unprocessed, does not directly enter the initial CNN block. Without this block, the raw signal would be transferred from one unit to another through the shortcut paths, bypassing any form of normalization or scaling.

3) **Amount of Channels**: A dynamic approach is taken with regards to the number of channels across the five blocks. The count of feature maps is progressively increased from block to block within the first half of the CNN module and subsequently decreased. This design is based on the idea that the CNN initially extracts an expanding set of features from the input, aiming to enrich the contained information. Subsequently, the knowledge acquired is condensed to facilitate management by the subsequent BiGRU.

4) **Batch Normalization**: To enhance the training effectiveness, including convergence speed and generalization capacity of the model, we introduce normalization layers following each convolutional layer. In line with the prevailing literature, we adopt Batch Normalization (BN) as the chosen normalization technique, although we also conducted experiments with other forms of normalization layers, such as instance and layer normalization.

*2) Details Regarding Our Employed Multi-head Attention Module:* For enhanced comprehension, Fig. 3 and Fig. 4 provide visual representations of our multi-head attention (MHA) module. Notably, we adapt the MHA framework that was introduced by Vaswani et al. in their seminal work about the Transformer architecture [69] to our specific use case. Differing from the self-attention used in Transformers [69], the query $q$ is initialized as a random vector and jointly learned. Moreover, we apply a $tanh$ activation function after each head's linear key transformation through $W_i^{(K)}$ to maintain the non-linearity within key derivation, similar to the attention mechanism in our baseline (cf. Fig. 2).

As in the main paper, Fig. 4 illustrates the internal mechanics of the scaled dot-product attention with Entmax15 activation in combination with multiple heads. Apart from the projection part, for which each head uses its own set of transformation matrices, the computations applied to determine the scaled dot-production are the same for each head, which is indicated by the layered shapes within the green box.

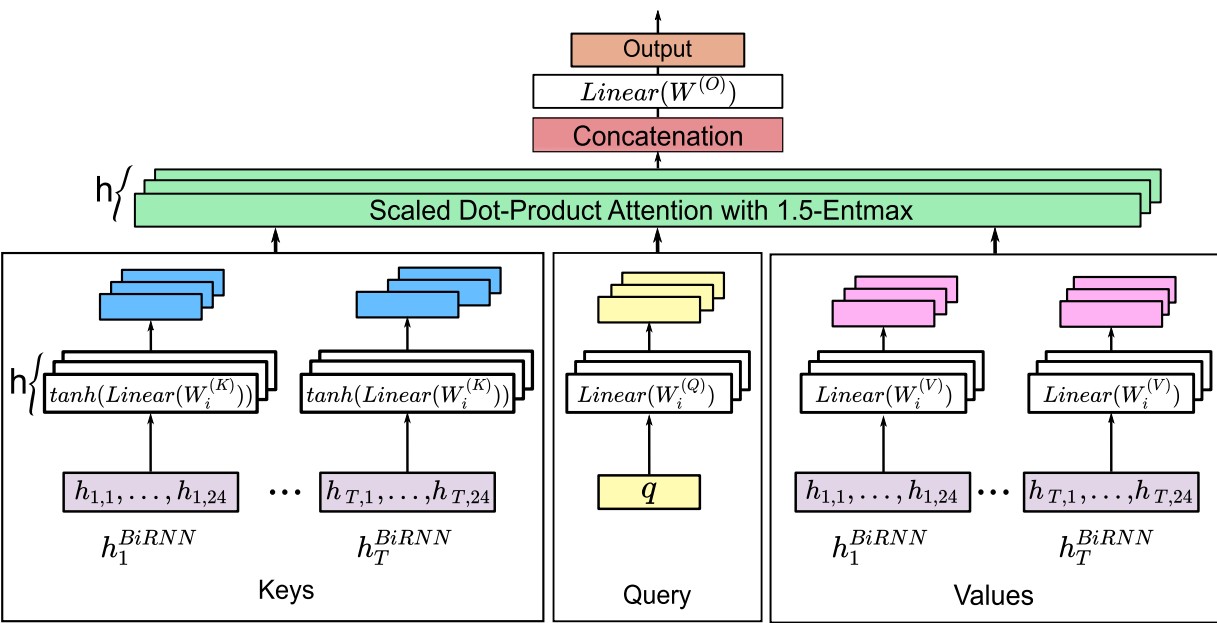

Fig. 3: Application of the MHA mechanism to the hidden states of a BiRNN with 12 units.

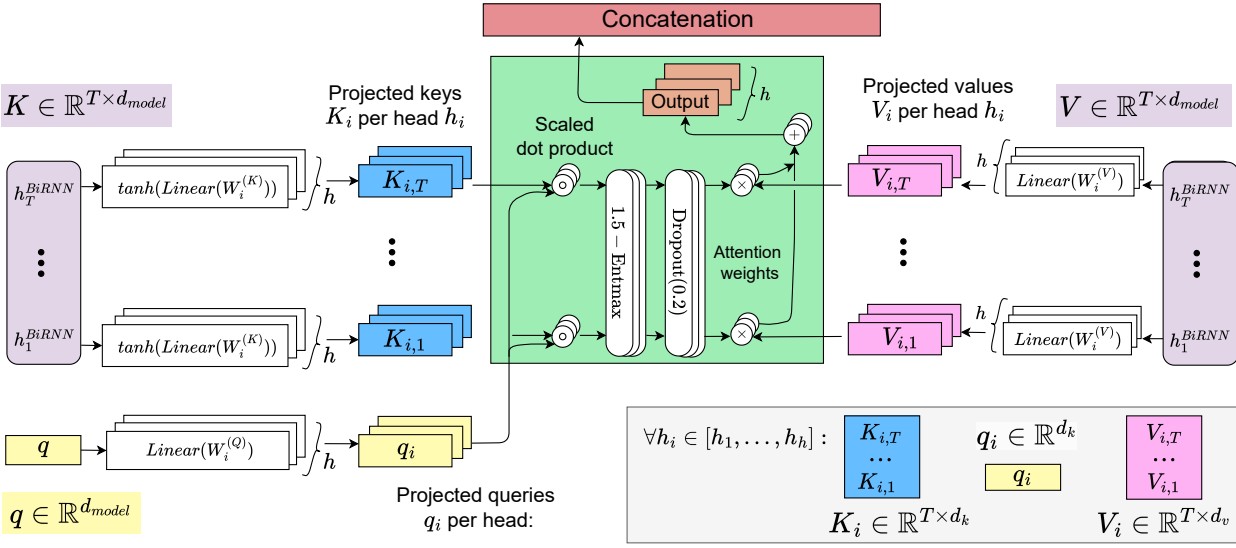

Fig. 4: Visualization of the scaled dot-product attention with $\alpha - Entmax$ and multiple heads from a zoomed-in perspective.

## III. DATASET AND DATA HANDLING

This section provides additional information about the CPSC benchmark dataset used for model evaluation in Subsection III-A. Following this, Subsection III-B presents details about our data splits during hyperparameter tuning and 10-fold cross-validation.

### A. Details Regarding the CPSC2018 Dataset

Our architectures are evaluated using the China Physiological Signal Challenge (CPSC) 2018 dataset. Before the competition, the dataset was split into two subsets with similar compositions. The first subset, with 6,877 records (3,178 female, 3,699 male), was the publicly available training data. The second subset, with 2,954 records (1,416 female, 1,538 male), was reserved for private evaluation and remains inaccessible to researchers. Hence, only the public data is used for training, validating, and evaluating our models.

As depicted on the left of Fig. 5, the majority of the 6,877 publicly available records have durations between six to 60 seconds. Nevertheless, there are exceptions, with 27 records lasting longer, reaching up to 144 seconds. On the right side of Fig. 5, the shares of the nine classes are summarized according to their first label annotations. These classes include normal sinus rhythm (SNR), as well as various cardiac disorders such as atrial fibrillation (AF), first-degree atrioventricular block (I-AVB), left bundle branch block (LBBB), right bundle branch block (RBBB), premature atrial contraction (PAC), premature ventricular contraction (PVC), ST-segment depression (STD), and ST-segment elevation (STE). Considering all label annotations, the shares increase for all classes except SNR, as shown ins Table IV.

### B. Details Regarding Our Data Splits During hyperparameter Tuning and 10-Fold Cross-Validation

For preliminary experiments and hyperparameter tuning, we used a fixed split of the 6,877 samples from the CPSC 2018 dataset. Specifically, 60% of the samples were used for training, while 20% each were used for validation and testing. Table VI provides the support of different classes within the three sets. To account for the lack of cross-validation in these early studies, we intentionally kept the training ratio relatively small. On one hand, this helps prevent overfitting on the training set. On the other hand, it increases the likelihood of covering a wider range of arrhythmia manifestations during evaluation.

For the main evaluation, we used 10-fold cross-validation with ten randomly partitioned folds of similar compositions. Each round involved one fold as the validation set, one as the test set, and the remaining eight for training. This process was repeated ten times, resulting in evaluations of the model on 10 distinct test sets with similar compositions. Table VII displays the support of different classes in the ten folds, each taking a turn as the unseen test set once.

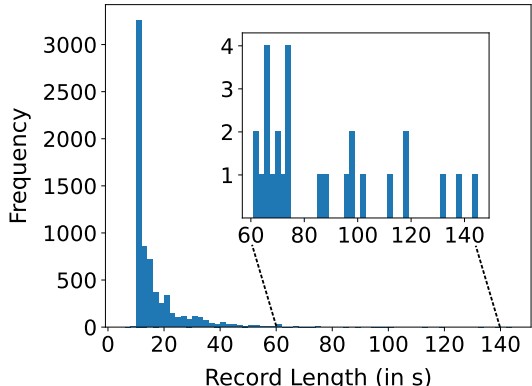

Histogram of record lengths.

| Type | Records |
|---|---|
| Normal (SNR) | 918 |
| Atrial fibrillation (AF) | 1098 |
| First-degree atrioventricular block(I-AVB) | 704 |
| Left bundle branch block (LBBB) | 207 |
| Right bundle branch block (RBBB) | 1695 |
| Premature atrial contraction (PAC) | 574 |
| Premature ventricular contraction (PVC) | 653 |
| ST-segment depression (STD) | 826 |
| ST-segment elevated (STE) | 202 |
| **Total** | 6877 |

Class shares.

Fig. 5: Details about the 6,877 records of the CPSC 2018

TABLE IV: Class shares when considering only the first vs. all annotations

| Mode | Classes | | | | | | | | | Total |
|---|---|---|---|---|---|---|---|---|---|---|
| | SNR | AF | I-AVB | LBBB | RBBB | PAC | PVC | STD | STE | |
| **1st label** | 918 | 1098 | 704 | 207 | 1695 | 574 | 653 | 826 | 202 | 6877 |
| **Multi-label** | 918 | 1221 | 722 | 236 | 1857 | 616 | 700 | 869 | 220 | 7359 |

TABLE V: Breakdown of class distribution - number of occurrences per class in total, in the multiple labeled (ML) records and as a percentage of multiple labeled in relation to all records.

| Metric | SNR | AF | I-AVB | LBBB | RBBB | PAC | PVC | STD | STE |
|---|---|---|---|---|---|---|---|---|---|
| **N (Total)** | 918 | 1221 | 722 | 236 | 1857 | 616 | 700 | 869 | 220 |
| **N (in ML records)** | 0 | 245 | 36 | 57 | 324 | 83 | 93 | 85 | 35 |
| **Share in %** | 0.00 | 20.07 | 4.99 | 24.15 | 17.4 | 13.47 | 13.29 | 9.78 | 15.91 |

TABLE VI: Support of different classes during preliminary experiments and hyperparameter tuning. Reported numbers are the same for all studies.

| Set | Classes | | | | | | | | | Total |
|---|---|---|---|---|---|---|---|---|---|---|
| | SNR | AF | I-AVB | LBBB | RBBB | PAC | PVC | STD | STE | |
| Train | 523 | 741 | 456 | 131 | 1091 | 379 | 421 | 528 | 131 | 4401 |
| Valid | 203 | 230 | 125 | 58 | 386 | 124 | 136 | 174 | 41 | 1477 |
| Test | 192 | 250 | 141 | 47 | 380 | 113 | 143 | 167 | 48 | 1481 |
| **Total** | 918 | 1221 | 722 | 236 | 1857 | 616 | 700 | 869 | 220 | 7359 |

TABLE VII: Support of different classes within the distinct test folds of the 10-fold cross-validation. Reported numbers are the same for all cross-validation studies.

| ID | Classes | | | | | | | | | Total |
|---|---|---|---|---|---|---|---|---|---|---|
| | SNR | AF | I-AVB | LBBB | RBBB | PAC | PVC | STD | STE | |
| 1 | 99 | 116 | 73 | 26 | 187 | 55 | 74 | 98 | 20 | 748 |
| 2 | 72 | 120 | 67 | 34 | 177 | 73 | 69 | 97 | 23 | 732 |
| 3 | 102 | 121 | 87 | 19 | 184 | 56 | 65 | 78 | 18 | 730 |
| 4 | 95 | 119 | 73 | 20 | 179 | 57 | 77 | 84 | 25 | 729 |
| 5 | 93 | 133 | 60 | 22 | 178 | 75 | 69 | 91 | 22 | 743 |
| 6 | 94 | 126 | 67 | 26 | 193 | 64 | 63 | 89 | 24 | 746 |
| 7 | 100 | 121 | 71 | 22 | 190 | 63 | 62 | 82 | 20 | 731 |
| 8 | 94 | 125 | 73 | 23 | 185 | 61 | 66 | 84 | 21 | 732 |
| 9 | 87 | 117 | 79 | 28 | 195 | 55 | 78 | 72 | 22 | 733 |
| 10 | 82 | 123 | 72 | 16 | 189 | 57 | 77 | 94 | 25 | 735 |
| **Total** | 918 | 1221 | 722 | 236 | 1857 | 616 | 700 | 869 | 220 | 7359 |

## IV. Evaluation

This section provides supplementary material on our approach evaluation. It includes visual performance representations of our models across nine classes (Subsection IV-A), feature importance from gradient boosting models (Subsection IV-B), and sigmoid activation visualizations for various classes and models (Subsection IV-C). It also details comparisons with state-of-the-art methods (Subsection IV-E) and a breakdown of our models' trainable parameters compared to existing approaches (Subsection IV-F).

### A. Receiver Operating Characteristic (ROC) Curves

Receiver Operating Characteristic (ROC) curves provide a clear visual representation of the performance of our models across the nine classes. We created a separate ROC plot for each class, including individual curves for each of the 10 folds from cross-validation, along with the mean curve computed across these 10 folds, for both Multi-Branch MACRO (MB-M) and the variant with MB-M as a feature extractor and subsequent GB classifiers based on all 117 input features (GB-all). The area shaded around the mean curve, which indicates plus or minus one standard deviation, provides a measure of the models' stability and robustness. The diagonal dashed line, indicative of chance-level performance, offers a baseline for comparison. These ROC plots, condensed into a grid with nine subplots for individual classes, are shown in Fig. 7 for MB-M and in Fig. 8 for MB-M with ensuing GB-all classifiers. Combined with the corresponding Area Under the Curve (AUC) scores, reported alongside, the plots provide a nuanced view of the model's discriminatory ability within each class and capture the consistency of performance across different folds.

### B. Feature Importance

In addition to evaluating model performance, we visualized the feature importance of gradient boosting (GB) models with different feature sets averaged across 10 folds in Fig. 6. These heatmaps offer valuable insights into the features that contribute most significantly to the classification of cardiac abnormalities. By analyzing the feature importance scores, one can gain a deeper understanding of the lead-specific branches that are indicative for different cardiac conditions.

The heatmap in Fig. 6a displays the importance of features in GB classifiers with 13 input features, namely the predicted probabilities from the BranchNets (BN) and Multi-Branch MACRO (MB-M) output layer for the respective class of interest. The most crucial feature for prediction is the MB-M output layer. However, for certain classes, such as bundle brunch blocks (LBBB and RBBB), specific BranchNet features also appear to play at least a minor role. The second heatmap in Fig. 6b shows the feature importance for GB classifiers with input 12 features, namely the predicted probabilities from the BranchNets only, excluding MB-M. No clear overall trend is discernible, but certain branches appear to be considered more frequently for certain classes than others. When using MB-M in combination with GB-all classifiers that take the 117

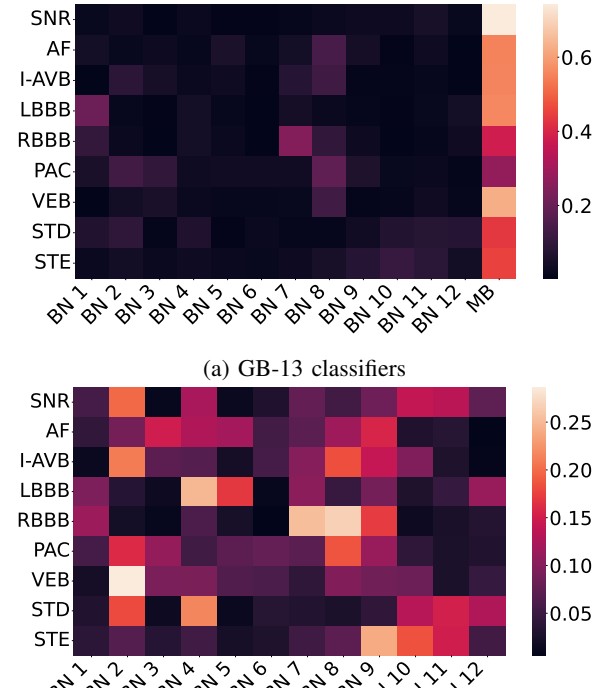

(a) GB-13 classifiers

(b) GB-12 classifiers

Fig. 6: Average feature importance across the 10 folds.

predicted probabilities from both the BranchNets and MB-M as input, the resulting heatmap is similar to that of 6a, i.e., the predicted probability output by MB-M for the current class of interest appears to be the most important, regardless of the specific class that the GB model is focusing on.

### C. Visualization of Final Sigmoid Probabilities

To enhance the interpretability of our approach, we present a visual representation of the final sigmoid probabilities of each classifier for all 12 BranchNets, MB-M, and its extensions with GB-13 and GB-all classifiers. To this end, the activations for ECG records where the class of interest is active (label = 1) are filtered and the mean activation values per class are calculated. Two distinct visualization strategies are employed. The first strategy entails plotting activations per class, thereby facilitating a comparison of model outputs for each specific class (cf. Fig. 9). The second strategy involves plotting activations per classifier, which allows for the visualization of the performance of each model across classes (cf. Fig. 10).

The viszalizations indicate that for the majority of classes, the highest activation occurs at the corresponding class index, thereby signifying accurate recognition with low probabilities assigned to other classes. However, the BranchNets exhibit a tendency to confuse the minority class STE with the SNR class (normal sinus rhythm). The best recognition for STE is still achieved with those based on leads V3 and V4. In contrast, the MB-M and GB variants improve recognition by a large margin, shifting the peak towards the correct prediction and reducing the likelihood of confusion with SNR.

Upon analysis of the results, it can be observed that, with the exception of STE, each BranchNet demonstrates an ability to accurately identify the corresponding cardiac disorder, even when utilizing a single lead as the input. However, for certain classes, such as LBBB or STD, occasional activations for other classes, such as RBBB or SNR, can also occur, albeit with reduced confidence. Moreover, the combination of all Branch-Nets in the shape of MB-M and the addition of GB classifiers has the effect of stabilizing the predictions, increasing confidence in the correct class while reducing it for others. While this improvement is particularly evident for the STE minority class, which aligns with the observed boost in performance metrics for STE, this general trend across classes indicates that merging predictions across multiple classifiers is beneficial for achieving more robust outcomes. However, it is important to note that this interpretation should be considered with caution, as a small number of records are multi-labeled. In such cases, co-activations for other classes might be appropriate, rather than indicative of misclassification.

### D. Comparison of Classification Performance for Single- and Multi-Labeled Records

While the CPSC 2018 dataset is primarily single-labeled, 476 records have more than one associated class. Table VIII divides the 6,877 records into single- and multi-labeled, showing separate performance metrics for both groups. Especially for MB-M, classification performance decreases from single-labeled (SL) to multi-labeled (ML) settings for all metrics except the precision, highlighting the increased complexity of multi-label classification. In contrast, the gradient boosting (GB) classifiers tend to handle the multi-label scenario more effectively than MB-M, reducing the difference in performance across all metrics. Although the AUC scores for the ML subset fall short of those for its SL counterpart, the results for both GB variants show that there are hardly any differences in the macro $F_1$ and accuracy scores. In some cases, the values achieved for SL are even exceeded for the ML category. Overall, the gradient boosting models (GB-all and GB-13) show improved performance in the multi-labeled context compared to MB-M, especially in terms of recall and $F_1$ score. This suggests that the binary ensemble of GB classifiers might be better suited for handling the ML records of CPSC 2018.

TABLE VIII: Comparison of the (macro-averaged) classification performance between single- and multi-labeled records.

| Model | Type | Prec. | Rec. | $F_1$ | AUC | Acc |
|---|---|---|---|---|---|---|
| **MB-M** | SL | 0.8304 | 0.8353 | 0.8318 | 0.9762 | 0.9704 |
| **MB-M** | ML | 0.8676 | 0.5945 | 0.6811 | 0.8534 | 0.9477 |
| **+GB-13** | SL | 0.8401 | 0.8354 | 0.8374 | 0.9661 | 0.9713 |
| **+GB-13** | ML | 0.8841 | 0.7808 | 0.8282 | 0.8727 | 0.9809 |
| **+GB-all** | SL | 0.8361 | 0.8379 | 0.8368 | 0.9651 | 0.9714 |
| **+GB-all** | ML | 0.8889 | 0.8127 | 0.8483 | 0.8671 | 0.9879 |

### E. Comparison to Existing SOTA Techniques

*1) Selection Process:* As stated in the main article, we excluded several studies from our comparison despite their analysis using CPSC 2018, as they were not comparable to our approach. Evaluation setups that differ from ours and were therefore excluded include the use of a fixed data split without k-fold cross-validation (CV) [60], [70]–[72], using a fixed test split combined with a (repeated) CV only for the training and validation sets [73]–[75], or using CV but reporting metrics of the best fold instead of averaging over all folds [40].

Although, in our tables, we report the results as they are given in the respective publications, for MLBF-Net [57] we made intensive efforts to reproduce the results ourselves due to its similarity to MB-M. However, despite our best efforts, we had great difficulty reproducing the results of the original publication. Since attempts to contact the authors were unsuccessful, and since other authors have also reported certain reproducibility problems in the meantime [76], we excluded MLBF-Net from our comparison, despite its 10-fold CV.

*2) Details:* Although we only included approaches in our SOTA comparison that have applied 5- or 10-fold CV scheme similar to ours, the majority of existing methods only reports average metric scores. In contrast, standard deviations across different folds are discussed in the least publications, despite providing valuable insights. In order to take this into account, Table IX takes up all approaches from the main paper that also report standard deviations and compares them with the results of our Multi-Branch MACRO (MB-M) with and without subsequent gradient boosting (GB) classifiers.

### F. Amount of Parameters

Our paper emphasizes transparency by providing comprehensive details on model architectures and parameter counts. The number of trainable parameters for our models is broken down in detail in Tables XI and XII for MACRO and Multi-Branch MACRO (MB-M), respectively. Table X compares the parameter counts of our models with those reported in other papers from our SOTA comparison (cf. Table 2 in the main paper), where possible. Publications that do not provide code or information on parameter counts [58], [62], [63] are excluded. The achieved macro $F_1$ (m-$F_1$) scores on the CPSC2018 dataset are included in the table as well, providing a comprehensive insight into the relationship between model complexity and performance across methods. If the original publication provides the number of parameters, we use this value in the table. For publications that have published PyTorch source code, we determined the number of parameters ourselves using the 'summary' method of the Python package 'torchinfo'[1]. This method was also used to count the parameters of MACRO and MB-M, ensuring consistent results. For source code using Keras, we employed the 'summary' method of Keras' models[2]. As shown in Table X, our models stand out by a thoughtful balance between model complexity and computational efficiency on the one hand, and classification performance on the other.

---

[1]PyTorch package 'torchinfo': https://github.com/TylerYep/torchinfo
[2]Keras method 'summary': https://keras.io/api/models/model/#summary-method

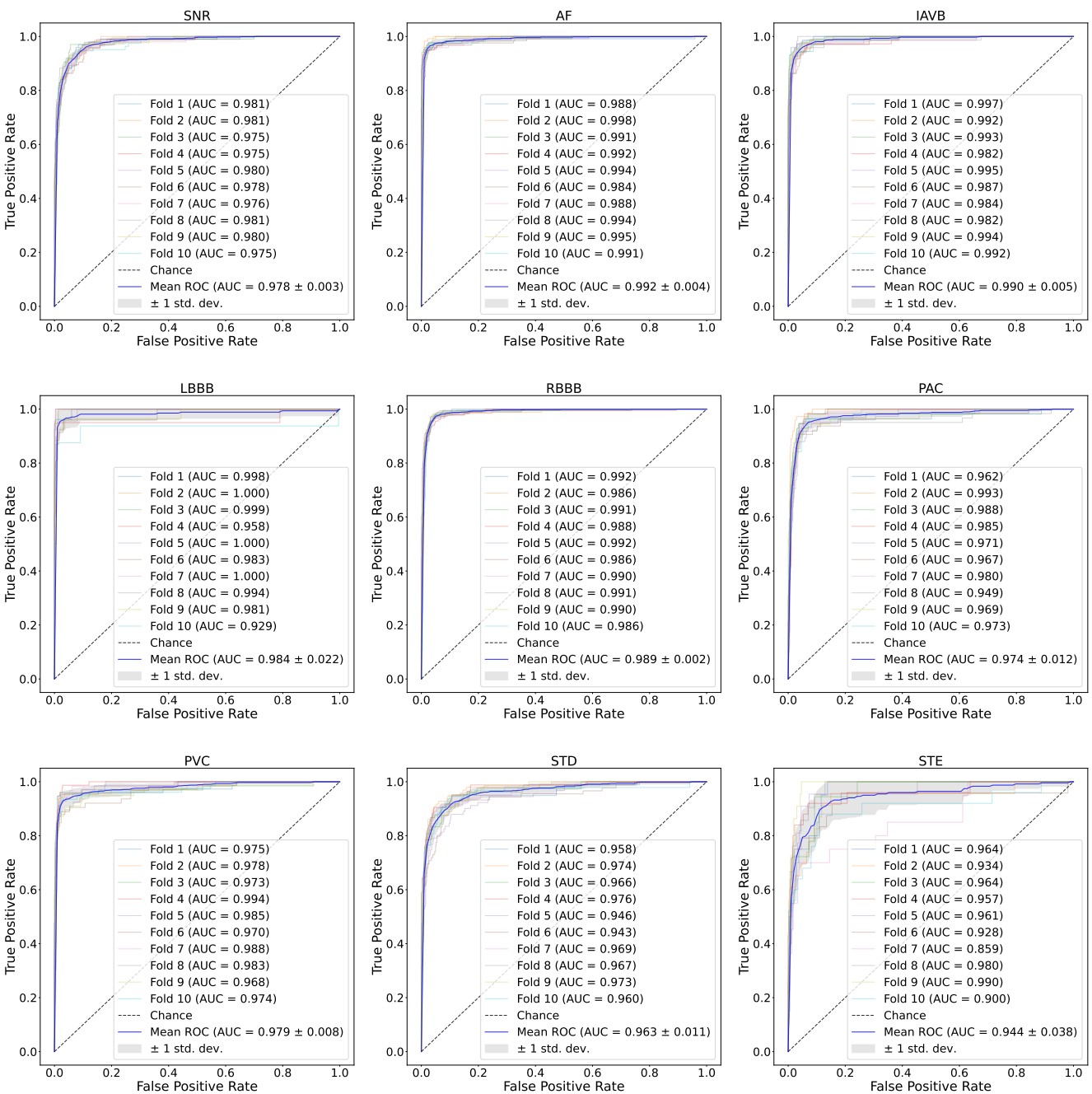

Fig. 7: Visualization of the ROC curves of our Multi-Branch MACRO model without gradient boosting (GB) classifiers across the 10 folds.

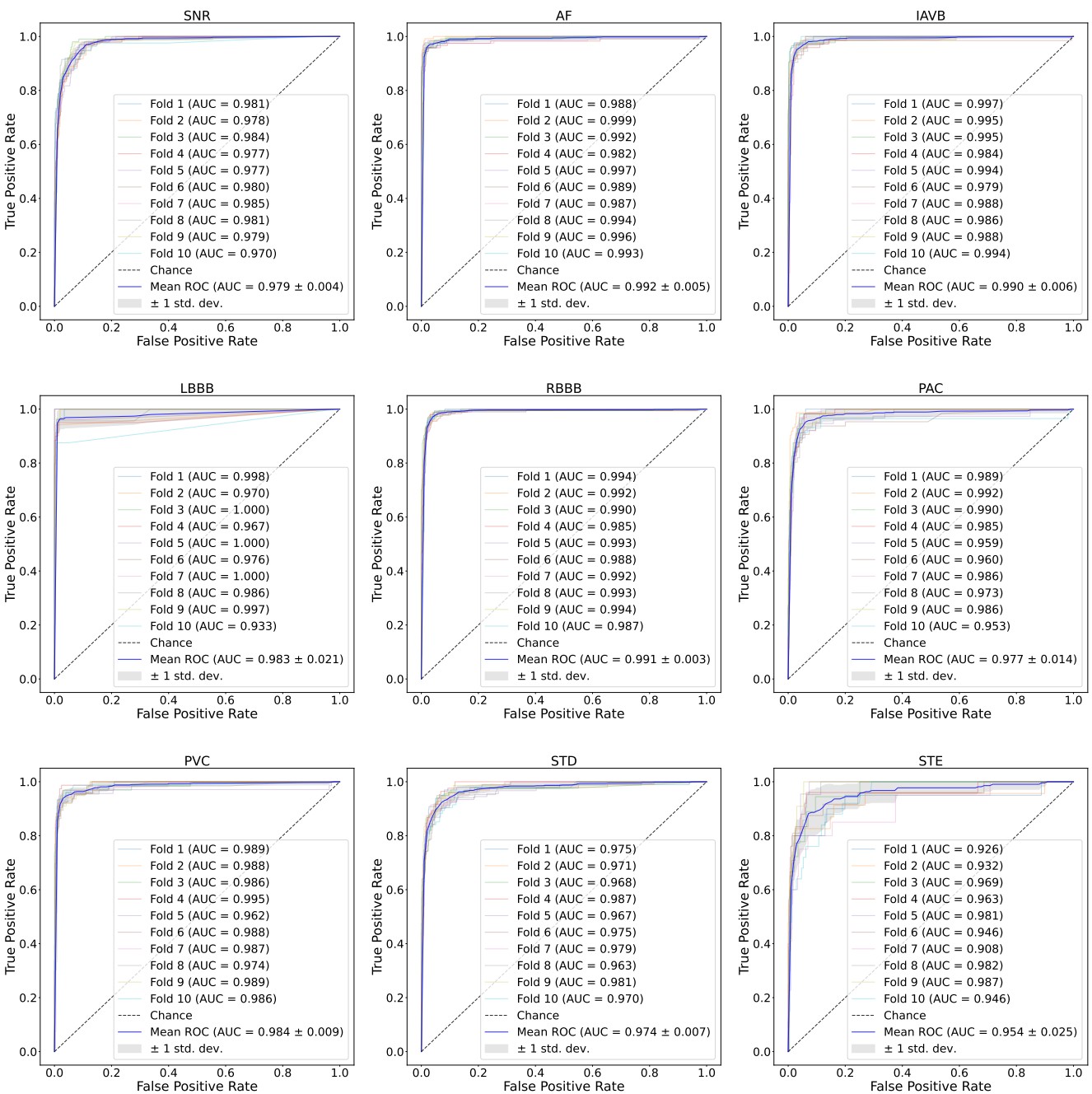

Fig. 8: Visualization of the ROC curves of our Multi-Branch MACRO model with subsequent GB-all classifiers across the 10 folds.

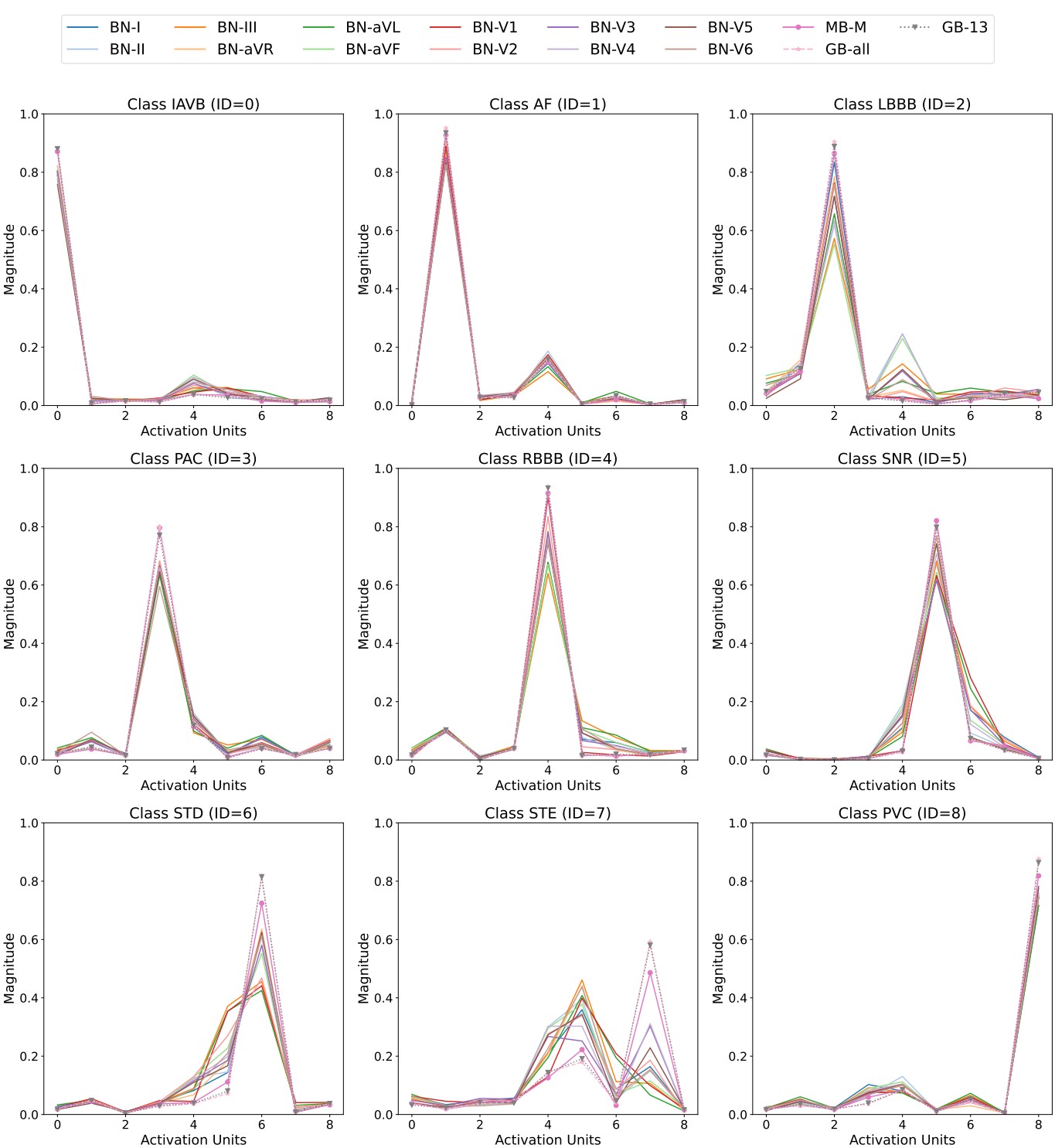

Fig. 9: Visualization of the final sigmoid activations of the different BranchNets, MB-M, and MB-M combined with GB-13 or GB-all, grouped by class.

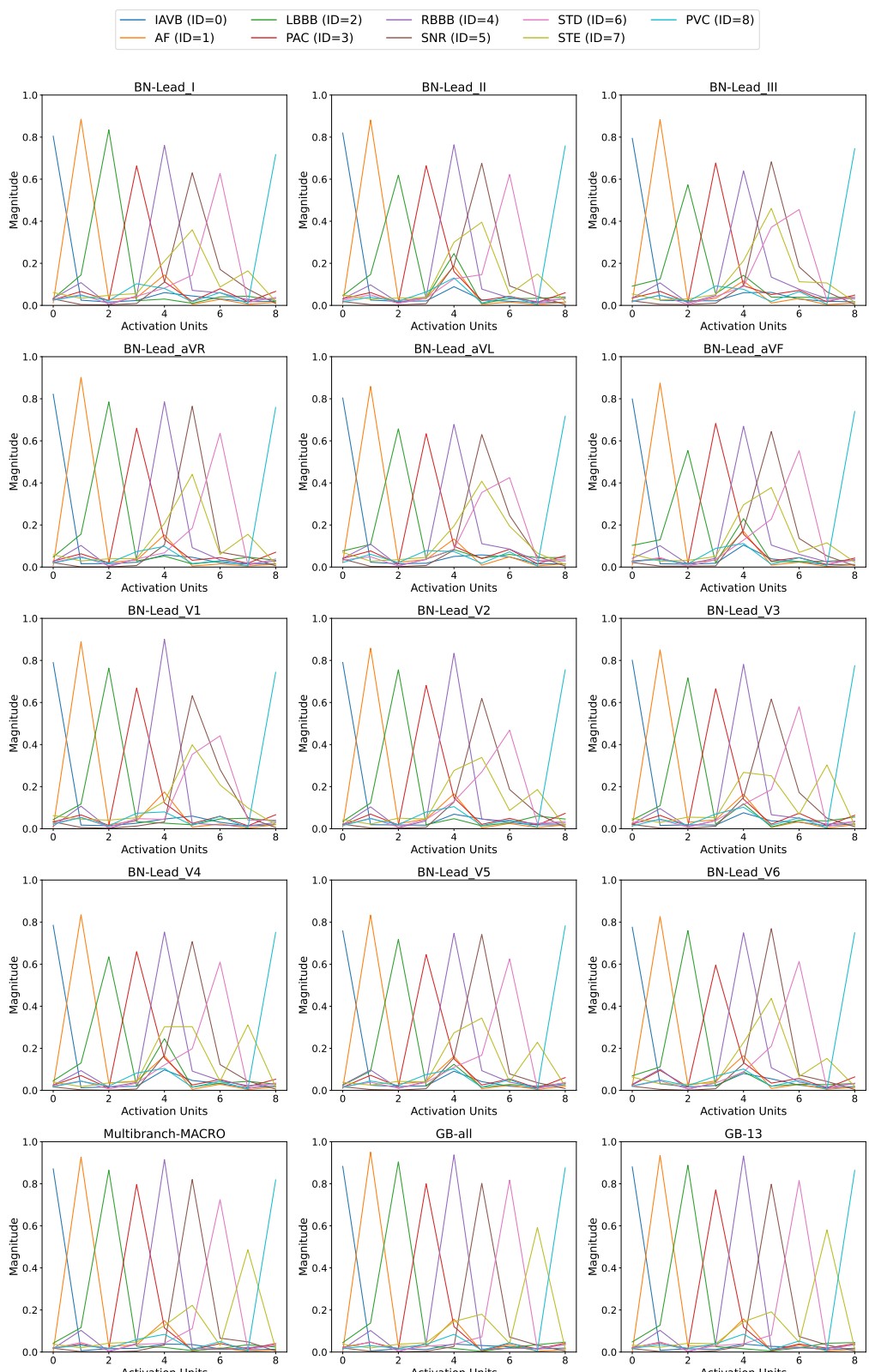

Fig. 10: Visualization of the final sigmoid activations of the different BranchNets, MB-M, and MB-M combined with GB-13 or GB-all, grouped by classifier.

TABLE IX: Model comparison regarding the class-wise and macro-averaged $F_1$ (m-$F_1$) scores achieved by our approach and existing SOTA methods (mean±sd).

| Type | ResCNN + BiLSTM [46] | Conv. SENet [48] | MsgFormer [58] | Multi-branch MACRO | | |
| --- | --- | --- | --- | --- | --- | --- |
| | | | | w/o GB | GB (13 ft.) | GB (all) |
| SNR | 0.755 ± 0.049 | 0.79 ± 0.03 | **0.840** ± 0.05 | 0.828±0.016 | 0.830±0.019 | 0.831±0.026 |
| AF | 0.846 ± 0.024 | 0.92 ± 0.02 | 0.923 ± 0.017 | 0.937±0.014 | 0.941±0.013 | **0.946**±0.011 |
| I-AVB | 0.87 ± 0.021 | 0.87 ± 0.04 | 0.838 ± 0.042 | 0.886±0.036 | **0.892**±0.036 | 0.888±0.036 |
| LBBB | 0.869 ± 0.028 | 0.87 ± 0.04 | 0.849 ± 0.021 | 0.866±0.057 | 0.896±0.051 | **0.903**±0.064 |
| RBBB | 0.78 ± 0.028 | 0.93 ± 0.01 | 0.935 ± 0.005 | 0.929±0.008 | 0.938±0.010 | **0.939**±0.008 |
| PAC | 0.751 ± 0.03 | 0.78 ± 0.05 | 0.731 ± 0.069 | 0.792±0.040 | 0.791±0.047 | **0.814**±0.038 |
| PVC | 0.829 ± 0.019 | 0.86 ± 0.03 | 0.856 ± 0.046 | 0.860±0.021 | 0.878±0.020 | **0.880**±0.019 |
| STD | 0.791 ± 0.01 | 0.81 ± 0.03 | **0.856** ± 0.014 | 0.790±0.034 | 0.832±0.034 | 0.834±0.026 |
| STE | **0.704** ± 0.049 | 0.59 ± 0.10 | 0.598 ± 0.052 | 0.557±0.062 | 0.643±0.090 | 0.630±0.051 |
| m-$F_1$ | 0.799 ± 0.014 | 0.825 ± 0.01 | 0.847 ± 0.013 | 0.827±0.014 | 0.849±0.012 | **0.852**±0.012 |

TABLE X: Comparison of the number of trainable parameters of MACRO and Multi-Branch MACRO with those employed in existing methods.

| Ref | Year | Model | Parameters | | m-$F_1$ | Details |
| --- | --- | --- | --- | --- | --- | --- |
| | | | Total | Rounded | | |
| [46] | '19 | ResCNN, BiLSTM | 1,163,913 | 1.16M [3] | 0.799 | |
| [65] | '23 | LightX3ECG | 5,343,732 | 5.34M [2] | 0.800 | Params. w/o pruning |
| [44] | '21 | CNN + Thld. Opt. | 16,610,185 | 16.61M [2] | 0.813 | ResNet34 (def.): 16.61M ResNet18: 8.75M |
| [56] | '23 | STRL + ASTA [4] | 2,419,542 | 2.42M [1] | 0.818 | |
| [48] | '21 | Conv. SENet | - | ≈3.5M [1] | 0.825 | |
| [64] | '24 | LFG-Net | - | 1.02M [1] | 0.842 | 1.02M only for prediction[5] |
| MACRO | | | 191,913 | **0.19M** | 0.809 | |
| MB-M | | | 1,713,081 | 1.71M | 0.827 | |
| MB-M + GB-all | | | 1,713,081 | 1.71M | **0.852** | No add. param. due to GB |

[1] Numbers as reported in the orig. publication
[2] Numbers determined with 'summary' method of the 'torchinfo' package
[3] Numbers determined with 'summary' method of Keras' models
[4] Spatio-Temporal Representation Learning + Attentive Spatio-Temporal Aggregation
[5] Training requires another model of 12 parallel networks, where each network structure contains CNN, BiGRU, attention, and fully-connected layers

TABLE XI: Specification of the trainable weight parameters of MACRO when trained with 12-lead ECG inputs and not acting as BranchNet, including the majority of selected hyperparameters.

| Layer set | Layers | #Kernels/units | K/H | Parameters | |
|---|---|---|---|---|---|
| Up-front block | Conv | 12 | 16 | $12 \times 12 \times 16 + 12 = 2,316$ | |
| | BatchNorm | - | - | $12 \times 2 = 24$ | |
| ResBlock 1 | Conv | 24 | 3 | $12 \times 24 \times 3 + 24 = 888$ | |
| | BatchNorm | - | - | $24 \times 2 = 48$ | |
| | Conv | 24 | 3 | $24 \times 24 \times 3 + 24 = 1,752$ | |
| | BatchNorm | - | - | $24 \times 2 = 48$ | $\sum = 16,920$ |
| | Conv | 24 | 24 | $24 \times 24 \times 24 + 24 = 13,848$ | |
| | ConvAlign | 24 | 1 | $12 \times 24 = 288$ (no bias) | |
| | BatchNorm | - | - | $24 \times 2 = 48$ | |
| ResBlock 2 | Conv | 48 | 3 | $24 \times 48 \times 3 + 48 = 3,504$ | |
| | BatchNorm | - | - | $48 \times 2 = 96$ | |
| | Conv | 48 | 3 | $48 \times 48 \times 3 + 48 = 6,960$ | |
| | BatchNorm | - | - | $48 \times 2 = 96$ | $\sum = 67,248$ |
| | Conv | 48 | 24 | $48 \times 48 \times 24 + 48 = 55,344$ | |
| | ConvAlign | 48 | 1 | $24 \times 48 = 1,152$ (no bias) | |
| | BatchNorm | - | - | $48 \times 2 = 96$ | |
| ResBlock 3 | Conv | 48 | 3 | $48 \times 48 \times 3 + 48 = 6,960$ | |
| | BatchNorm | - | - | $48 \times 2 = 96$ | |
| | Conv | 48 | 3 | $48 \times 48 \times 3 + 48 = 6,960$ | |
| | BatchNorm | - | - | $48 \times 2 = 96$ | $\sum = 71,856$ |
| | Conv | 48 | 24 | $48 \times 48 \times 24 + 48 = 55,344$ | |
| | ConvAlign | 48 | 1 | $48 \times 48 = 2,304$ (no bias) | |
| | BatchNorm | - | - | $48 \times 2 = 96$ | |
| ResBlock 4 | Conv | 24 | 3 | $48 \times 24 \times 3 + 24 = 3,480$ | |
| | BatchNorm | - | - | $24 \times 2 = 48$ | |
| | Conv | 24 | 3 | $24 \times 24 \times 3 + 24 = 1,752$ | |
| | BatchNorm | - | - | $24 \times 2 = 48$ | $\sum = 20,376$ |
| | Conv | 24 | 24 | $24 \times 24 \times 24 + 24 = 13,848$ | |
| | ConvAlign | 24 | 1 | $48 \times 24 = 1,152$ (no bias) | |
| | BatchNorm | - | - | $24 \times 2 = 48$ | |
| ResBlock 5 | Conv | 12 | 3 | $24 \times 12 \times 3 + 12 = 876$ | |
| | BatchNorm | - | - | $12 \times 2 = 24$ | |
| | Conv | 12 | 3 | $12 \times 12 \times 3 + 12 = 444$ | |
| | BatchNorm | - | - | $12 \times 2 = 24$ | $\sum = 8,604$ |
| | Conv | 12 | 48 | $12 \times 12 \times 48 + 12 = 6,924$ | |
| | ConvAlign | 12 | 1 | $24 \times 12 = 288$ (no bias) | |
| | BatchNorm | - | - | $12 \times 2 = 24$ | |
| BiGRU | | 12 | | $2 \times 3 \times (12^2 + 12 \times 12 + 2 \times 12) = 1,872$ | |
| MH attention | Learnable query vector | | | 24 | |
| | Dense (key transformations) | 6 | | $24 \times 24 + 24 = 600$ | |
| | Dense (query transformations) | 6 | | $24 \times 24 + 24 = 600$ | $\sum = 2,424$ |
| | Dense (value transformations) | 6 | | $24 \times 24 + 24 = 600$ | |
| | Dense (head fusion) | - | | $24 \times 24 + 24 = 600$ | |
| BatchNorm | - | - | - | $24 \times 2 = 48$ | $\sum = 273$ |
| Dense | - | - | - | $24 \times 9 + 9 = 225$ | |
| **Total** | | | | **191,913** | |

Number of parameters computed as follows, where # denotes the amount:
Conv: #inputChannels · #kernels · kernelSize+ #kernels (bias)
BatchNorm: #inputChannels · 2
BiGRU: #directions · #gates · ((#cells)$^2$ + inputSize · #cells + 2 · #cells), cf. PyTorch doc.
Dense: inputSize · outputSize + outputSize (bias)
MH Transformations: like dense layer, where in this work outputSize = 2 · #cells$_{\text{BiGRU}}$

TABLE XII: Specification of the trainable weight parameters of Multi-Branch MACRO, including an overview of the majority of selected hyperparameters.

| Block | Layer (set) | Num. of Kernels | K | Parameters | | |
|---|---|---|---|---|---|---|
| BranchNet | Up-front block | Conv — 1 | 16 | 17 | } $\sum = 19$ | } |
| | | BatchNorm — - | - | 2 | | |
| | ResBlock1 | Conv — 12 | 3 | 48 | } $\sum = 4,044$ | |
| | | BatchNorm — - | - | 24 | | |
| | | Conv — 12 | 3 | 444 | | |
| | | BatchNorm — - | - | 24 | | |
| | | Conv — 12 | 24 | 3,468 | | |
| | | ConvAlign — 12 | 1 | 12 | | |
| | | BatchNorm — - | - | 24 | | $\times\,12$ |
| | ResBlock2 | cf 1st block of MACRO (in_ch: 12, out_ch: 24) | | 16,920 | } $\sum = 117,717$ | |
| | ResBlock3 | cf. 2nd block of MACRO (in_ch: 24, out_ch: 48) | | 67,248 | | |
| | ResBlock4 | cf. 4th block of MACRO (in_ch: 48, out_ch: 24) | | 20,376 | | |
| | ResBlock5 | cf. 5th block of MACRO (in_ch: 24, out_ch: 12) | | 8,604 | | |
| | BiGRU | cf. MACRO (12 cells) | | 1,872 | | |
| | MHA | cf. MACRO, but 24 heads | | 2,424 | | |
| | Classification | cf. MACRO | | 273 | | |
| Conv Red | Conv | 200 | 3 | 173,000 | } $\sum = 249,024$ | |
| | BatchNorm | - | - | 400 | | |
| | Conv | 112 | 3 | 67,312 | | |
| | BatchNorm | - | - | 224 | | |
| | Conv | 24 | 3 | 8,088 | | |
| MHA | | cf. MACRO (6 heads) | | 2,424 | | |
| BatchNorm | | cf. MACRO | | 48 | | |
| Dense | | cf. MACRO | | 225 | | |
| **Total** | | | | **1,713,081** | | |