# OpenReview forum: "MACRO: A Multi-Head Attentional Convolutional Recurrent Network for The Classification of Co-Occurring Diseases in 12-Lead ECGs"
_IEEE.org/EMBS/BHI/2024/Conference — IEEE BHI'24_

### Official Review · Reviewer_FEZF · 2024-08-10
**Lead-Specific ECG Analysis Shows Potential; Effective Architecture for ECG Classification**

**Overall Rating:** 5
**Confidence:** 4

**Other Quality Metrics:**

Clarity of Writing: Good
Clinical Significance: Fair
Methodological Novelty: Good
Experiments and Results: Good

**Questions For The Authors:**

1. Generalizability: Have you evaluated or plan to evaluate the model on additional datasets such as PTB-XL? suggestion: The paper could benefit from experimenting with additional datasets, as the GPU resource allows.
2. Model Interpretability: Can you provide more insights into how the lead-specific modeling can be interpreted and used in a clinical setting? suggestion: thorough clinical interpretation is always difficult, but the authors could perform some analysis based on the trained model to demonstrate the effectiveness of the feature extractor. (For example, the “signatures” of different CVD manifest in some ECG leads, which could be interpreted by varying activations in MACRO’s multi-branch architecture.)

**Strengths:**

* Lead-specific analysis: The multi-branch approach allows for tailored processing of each ECG lead and will potentially enhance diagnostic accuracy. The relevant design choices are nicely detailed in Section II.B of the paper.
* Parameter efficiency: The proposed models achieve competitive performance with significantly fewer parameters compared to state-of-the-art methods. The paper highlights, "Our architecture remains lightweight with approximately 1.7 million trainable parameters, which represents a reduction in the number of parameters of up to 90% compared to previous methods" (Abstract).
* Effective architecture for feature extraction: The integration of CNNs, BiGRUs, and multi-head attention addresses both spatial and temporal aspects of ECG data effectively.
* Open-source code implementation

**Summary Of The Paper:**

The paper proposes a novel deep learning architecture (MACRO) for the automated detection of cardiovascular diseases in 12-lead ECG data. The architecture integrates convolutional neural networks (CNNs), bidirectional gated recurrent units (BiGRUs), and a multi-head attention mechanism to handle the complexity of ECG signals. Additionally, the paper presents a multi-branch variant (MB-M) to process each lead independently before feature fusion. The model's performance is evaluated on the CPSC 2018 dataset, achieving superior macro F1 and AUC scores compared to existing state-of-the-art methods while maintaining a relatively small parameter count.

**Weaknesses:**

1. Generalizability: The evaluation is limited to the CPSC 2018 dataset. Testing on additional datasets (such as PTB-XL and other physionet datasets) is necessary to demonstrate broader applicability. The paper briefly mentions future plans to evaluate on PTB-XL but does not present current results.
2. Clinical Relevance / Interpretability: While the model shows technical advancements, the paper could benefit from a deeper discussion on the clinical relevance and potential impact on patient care. The proposed method implemented lead-specific modeling (in the first conv layers at least), and the paper stressed handling “co-occurring” CVDs, but the analysis in that regard is lacking.

---

### Official Review · Reviewer_Mcrc · 2024-08-11
**New deep learning architecture for detecting abnormal irregularities from ECG**

**Overall Rating:** 7
**Confidence:** 3

**Other Quality Metrics:**

(a) Clarity of writing: Excellent
(b) Clinical Significance: Excellent
(c) Methodological Novelty: Great
(d) Experiments and Results: Excellent

**Questions For The Authors:**

I am curious about the potential for transfer learning with this new architecture. Given the growing interest in transfer learning, do the authors have any thoughts or plans for extending their multi-branch framework in this direction?

**Strengths:**

1. Innovative Approach: The authors present a multi-branch framework and fusion models that address current challenges in ECG irregularity detection effectively.
2. Clear Model Description: The architecture is clearly described, with a well-explained rationale behind the design choices.
3. Robust Experimental Design: The experiments, particularly those assessing the performance of MACRO and BM-M models, are well-designed. I appreciated the detailed comparison across individual labels (e.g., SNR, AF), as it provides a deeper insight into the model's strengths and potential limitations.

**Summary Of The Paper:**

The authors introduce a novel deep-learning architecture designed to detect concurrent irregularities in 12-lead ECG data. Their approach, tested on the China Physiological Signal Challenge 2018 dataset, shows competitive performance compared to state-of-the-art and baseline methods.

**Weaknesses:**

1. Minor suggestions:
- While the authors mention the majority vs. minority class labels, it would enhance the interpretability of the results to include these specific numbers (N) in Table I or as an appendix.
- The paper notes that out of all ECG recordings from the CPSC 2018, only 476 were multi-labeled. It would be helpful to clarify how well the models captured the multi-labels per recording. I was also wondering if the minority classes like STE predominantly appeared in these multi-labeled recordings-- this might support the effectiveness of this architecture. In general, a discussion on whether there was a performance difference between multi-labeled and single-labeled recordings would be valuable.

---

### Official Review · Reviewer_ppW1 · 2024-08-12
**Review: MACRO: A Multi-Head Attentional Convolutional Recurrent Network for the Classification of Co-Occurring Diseases in 12-Lead ECGs**

**Overall Rating:** 7
**Confidence:** 4

**Other Quality Metrics:**

* Clarity of writing: Great
* Clinical significance: Great
* Methodological novelty: Good
* Experiments and results: Great

**Questions For The Authors:**

* Datasets with 12 lead ECG data are often collected in clinical settings. In other words, it is not possible to collect all 12 leads from wearable devices. It was mentioned multiple times in the paper that there is a need to control model complexity and the number of parameters. If the model is not designed for a wearable device or to run on an edge device - why the model complexity still matters Furthermore, in this case - how can we utilize the proposed model MB-M with more than 1 million parameters?
* It is mentioned that the majority of ECG signals had only one label. How was the distribution of the labels? Are there any rare classes in this dataset? there should be more discussion on the results of the proposed approach on the ECG data points with labels from the minority classes.
* How this model can be modified for a setting with ECG data without all 12 leads?

**Strengths:**

* The authors provide the results of a physiological signal challenge which facilitates the comparison between the state-of-the-art methods.
* The paper has the advantage of utilizing all 12 leads coupled with an attention mechanism to attend more to the leads which provides more context about certain cardiovascular conditions such as AFib as mentioned in the text.
* The paper provides details about the deep learning architecture as well as analyses of the number of parameters.

**Summary Of The Paper:**

Electrocardiogram (ECG) signals are among the most critical biosignals for heart health assessment and the early detection and monitoring of cardiovascular conditions. Typically, these signals are collected using 12 leads. This paper proposes a solution that combines machine learning and deep learning approaches for feature extraction and multilabel classification. The authors also provide an extensive analysis of the results from the China Physiological Signal Challenge (CPSC) 2018.

**Weaknesses:**

* It is mentioned that Machine learning methods “ require complex preprocessing and sophisticated feature engineering.” This is a claim that should be supported by some literature as there are many preprocessing approaches for machine learning methods that are daily straightforward to use and more importantly, they are interpretable. I recommend the authors to rephrase this claim to avoid confusion.
* One of the advantages of the proposed approach is to have an attention head for each ECG lead. There should be more discussion on the effect of the attention value on different cardiovascular diseases.

---

### Decision · Program_Chairs · 2024-09-23

Accept